# Cortical circuits for cross-modal generalization

Maëlle Guyoton ®[1,3], Giulio Matteucci ®[1,3], Charlie G. Foucher ®[1], Matthew P. Getz ®[2], Julijana Gjorgjieva ®[2] & Sami El-Boustani ®[1]✉

Adapting goal-directed behaviors to changing sensory conditions is a fundamental aspect of intelligence. The brain uses abstract representations of the environment to generalize learned associations across sensory modalities. The circuit organization that mediates such cross-modal generalizations remains, however, unknown. Here, we demonstrate that mice can bidirectionally generalize sensorimotor task rules between touch and vision by using abstract representations of peri-personal space within the cortex. Using large-scale mapping in the dorsal cortex at single-cell resolution, we discovered multimodal neurons with congruent spatial representations within multiple associative areas of the dorsal and ventral streams. Optogenetic sensory substitution and systematic silencing of these associative areas revealed that a single area in the dorsal stream is necessary and sufficient for cross-modal generalization. Our results identify and comprehensively describe a cortical circuit organization that underlies an essential cognitive function, providing a structural and functional basis for abstract reasoning in the mammalian brain.

Objects possess unique physical properties that are detected by different sensory organs. The brain seamlessly integrates these sensory inputs to create unified percepts and abstract representations of the environment, which are essential for generalizing behaviors to unfamiliar situations[1]. This phenomenon is especially pronounced in the peri-personal space, where visual and tactile inputs converge[2]. Indeed, an object's position and identity, initially discerned through touch in darkness, can immediately be recognized by sight in light. Cross-modal generalization—also called cross-modal transfer learning—describes the process by which recognition in one sensory modality enables the generalization of learned associations to others, a capability observed across diverse species[2–5], suggesting a common foundational circuit organization. Across the hierarchy of sensory systems, neural representations become increasingly invariant to low-level features, including the specific sensory modality of a stimulus, ultimately abstracted to form representations that encode perception in a modality-independent manner. While neuronal correlates of such abstract representations have been identified—from supramodal stimulus feature encoding in the rat posterior parietal cortex to "concept cells" in the human temporal lobe[6–9]—the cortical architecture enabling their use for generalized learning across sensory modalities remains to be elucidated. Mice have proven to be a valuable model for dissecting circuits responsible for multisensory integration and their role in goal-directed behaviors[10–13]. Mice rely on visuo-tactile inputs for behaviors such as gap crossing[14], navigation[15], and object recognition[6]. Because whiskers occupy a substantial portion of the visual field, both somatosensory and visual systems frequently receive correlated inputs (Fig. 1a). In the superior colliculus, multimodal representations of whisker and visual information possess topographically co-aligned functional maps[16,17], potentially enhancing reflexive behaviors like gaze and head orientation by increasing the salience of spatially congruent multisensory events[18]. Cortical circuits, critical for perception and goal-directed behaviors, also display multimodal visuo-tactile responses. In particular, associative areas within the posterior parietal cortex receive inputs from both the primary visual cortex (V1) and the primary somatosensory cortex (S1)[19]. However, the functional and anatomical organization of cortical areas dedicated to visuo-tactile processing, as well as their potential role in cross-modal generalization

[1]Department of Basic Neurosciences, Faculty of Medicine, University of Geneva, 1 Rue Michel-Servet, 1206 Geneva, Switzerland. [2]School of Life Sciences, Technical University of Munich, Maximus-von-Imhof-Forum 3, 85354 Freising, Germany. [3]These authors contributed equally: Maëlle Guyoton, Giulio Matteucci. ✉e-mail: sami.el-boustani@unige.ch

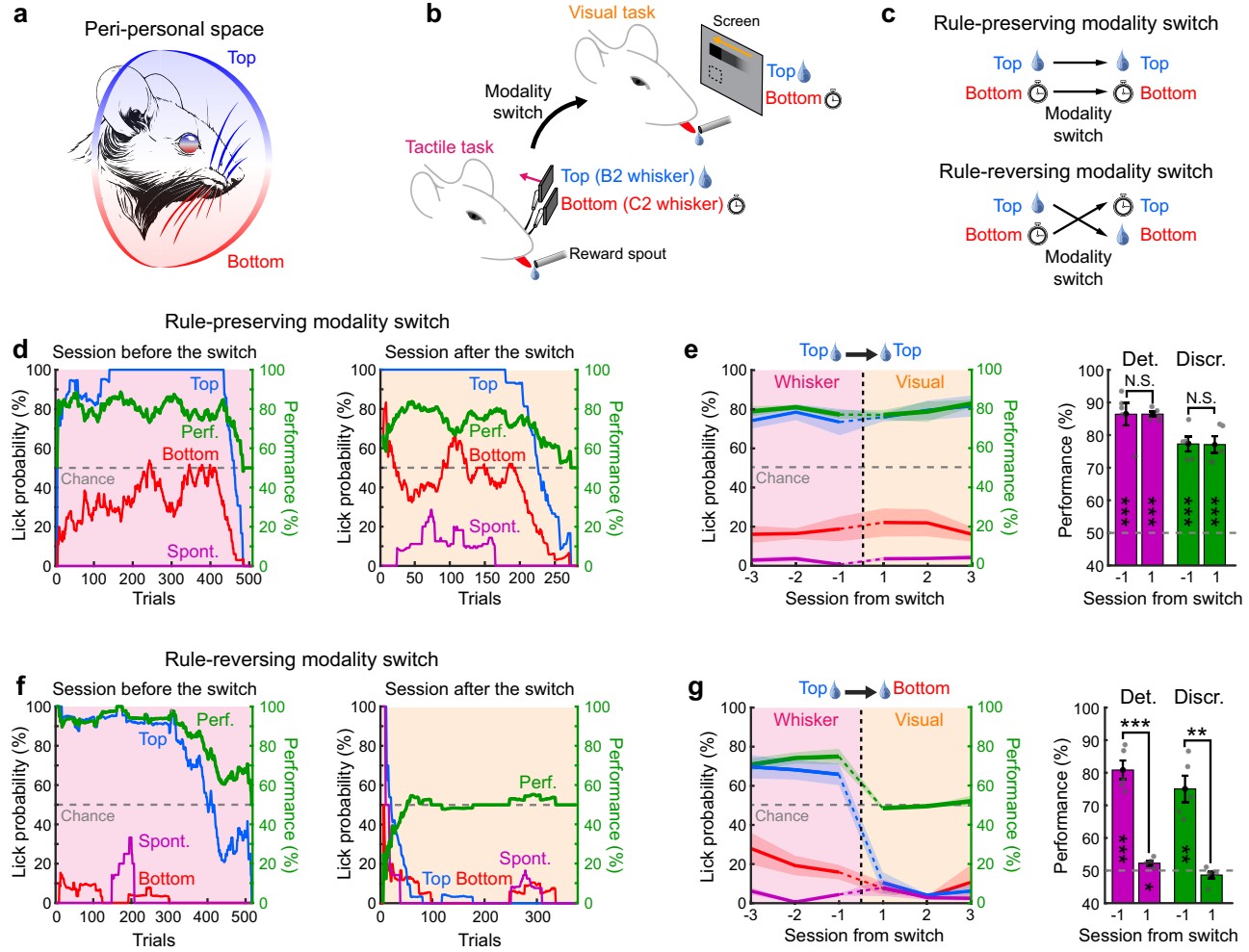

**Fig. 1 | Cross-modal generalization in mice. a** Illustration of the common organization of the peri-personal space for visual and whisker tactile inputs in mice. **b** Schematic of the behavioral Go/No go paradigm for studying cross-modal generalization of spatial information in mice. **c** Two types of tactile-to-visual modality switches: "rule-preserving", wherein the spatial location of rewarded stimuli is preserved, and "rule-reversing", wherein the location of rewarded stimuli is reversed. **d** Left: example of a session with the tactile task the day before modality switch. Conditional lick probabilities over trials are shown for the top whisker (blue), the bottom whisker (red) and in absence of stimuli (purple). Task performance (green) is computed as the percentage of correct discrimination trials (see "Methods"). Chance level is shown as a gray dashed line. Traces were computed on a sliding window of 60 trials. Right: same as left for the first visual session following a rule-preserving modality switch. **e** Left: task performance and conditional lick probabilities averaged across mice ($N = 5$ mice) over three consecutive sessions

before and after a rule-preserving switch (vertical dashed line). Shaded area: S.E.M. Color code as in panel d. Right: detection (purple) and discrimination (green) performance distribution for the session before and after the switch (two-sided paired $t$ test comparing days, Det.: N.S. $p = 1$; Discr.: N.S. $p = 0.96$). Performances are also tested against chance level (two-sided $t$ test, Det.: ***$p = 4.4 \times 10^{-4}$ and ***$p = 1.4 \times 10^{-7}$; Discr.: ***$p = 2.8 \times 10^{-4}$ and ***$p = 4.7 \times 10^{-4}$). Error bars: S.E.M. Discrimination performance indicates the proportion of trials in which mice correctly responded to top and bottom stimuli. Detection performance indicates the proportion of trials in which mice differentiated any stimulus (top or bottom) from no stimulus at all (see "Methods"). **f, g** Same as in panels d-e but for a rule-reversing modality switch (two-sided paired $t$ test comparing days, Det.: ***$p = 1.2 \times 10^{-5}$; Discr.: **$p = 0.005$). Performances are also tested against chance level (two-sided $t$ test, Det.: ***$p = 4.6 \times 10^{-4}$ and *$p = 0.03$; Discr.: **$p = 0.003$ and Blank $p = 0.29$).

of learned sensorimotor associations, remains unknown. Here, we show that mice rapidly generalize sensorimotor task rules between touch and vision by forming an abstract spatial representation of peripersonal space. Using wide-field and two-photon calcium imaging, anatomical tracing, and perturbative approaches, we find that a single area in the dorsal cortex is necessary and sufficient for cross-modal generalization. These results thus provide a detailed circuit mechanism and structural basis for how the mammalian brain abstracts and generalizes learned behaviors across sensory modalities.

## Results

### Cross-modal generalization in mice

We designed a behavioral paradigm to test the ability of mice to generalize sensorimotor associations learned through whisker sensations to the visual modality. Given that both sensory modalities share a

common spatial organization within the peri-personal space (Fig. 1a), the task includes visual and tactile stimuli that originate from locations that are spatially congruent (Fig. 1b). In the dark, head-fixed and water-restricted mice were first trained on a Go/No go tactile discrimination task, where they had to discriminate between stimulations of two whiskers vertically arranged in the same column of the whisker pad. Mice were rewarded with a drop of water if they licked a spout upon stimulation of the top whisker (B2) whereas they were punished with a 10-second-long timeout if they licked for the bottom whisker (C2). Once mice became expert at the task and performed stably with a high percentage of correct trials over at least 3 consecutive sessions, we switched the task to a Go/No go visual task. In this condition, we replaced the top and bottom whisker stimulations with top and bottom visual stimuli. These stimuli consisted of black squares on a gray background drifting along the same rostro-caudal direction than the

whisker deflections. The screen was oriented to be centered and parallel to the right retina on the same side as the previously stimulated whiskers. The location of the moving squares along the vertical axis was chosen to roughly match the location of the whiskers within the visual field of the animals (see "Methods").

To test whether mice can use previous associations learned during the tactile task to infer the reward contingency (i.e. the rule that determines when a reward will be delivered following a sensorimotor response) in a new visual task, we considered two scenarios: a "rule-preserving" and a "rule-reversing" modality switch. The cohort of mice undergoing the rule-preserving modality switch could obtain a reward by licking for stimuli presented at the same spatial location after the switch. For mice undergoing the rule-reversing switch, the reward contingency was spatially reversed following the switch to the new modality (Fig. 1c). Following rule-preserving modality switches, we observed rapid generalization of the learned association to the new modality. Mice seamlessly performed the task with a comparable level of high performance as observed in the session preceding the switch (Fig. 1d, e), already within the first few tens of trials. In contrast, task performance was strongly affected following a rule-reversing modality switch, consistently falling at chance level or below. Mice typically attempted to lick in response to both stimuli during the early part of the session before rapidly disengaging and ceasing any licking behavior (Fig. 1f). In many cases, mice attempted to lick first for visual stimuli spatially congruent with the previously rewarded whisker, causing performance to briefly drop below chance level during the early phase of the session. Following rule-reversing switches, mice displayed a strong resistance to engaging with the task for several consecutive sessions (Fig. 1g). We verified that this result was not caused by a preference for the top whisker or for the top visual stimulus by repeating the same experiments with cohorts of mice trained to lick for bottom whisker stimulations. Mice maintained high task performance after switches that preserved the spatial rule, regardless of whether they were initially trained on one whisker or another, but performance briefly dropped below chance when the rule was reversed ($N = 10$ mice, two-sided $t$ test with respect to chance level in the first 100 trials, $p = 0.02$), before eventually disengaging (Supplementary Fig. 1a–c). Such disengagement after a rule-reversing modality switch persisted despite strong thirst-driven motivation, as indicated by prolonged licking bouts observed when water drops were manually delivered to maintain task engagement[20]. To ensure comparable thirst levels across sessions, we carefully regulated water intake and monitored weight loss for each mouse. Furthermore, all cohorts were equally trained and exhibited comparable performance and engagement before the modality switch (Supplementary Fig. 2a–d). This implies that the mouse's failure to perform after a rule-reversing modality switch stems from conflicting prior knowledge of the task rule, rather than from a lack of experience or motivational drive.

Despite the limited visibility of the capillary glass tubes used for whisker stimulations in the dark, we tested whether mice could rely on the movement of tubes as visual cues to perform the tactile task, thereby generalizing within the visual domain instead of across sensory modalities. We carried out control experiments where mice proficient in the tactile task underwent sessions with whiskers temporarily removed from the tubes, and subsequently reintroduced. We observed that both detection and discrimination performance dropped to chance level immediately after the whiskers were removed from the capillary tubes, but recovered to expert levels once the whiskers were reinserted into the piezo stimulators (Supplementary Fig. 3). This demonstrates that mice were not using visual cues to perform the whisker discrimination task in the dark.

Besides generalizing task rule through common spatial organization, mice could also potentially generalize the abstract Go/No go structure common to the two tasks (i.e. "acting upon one stimulation, the Go stimulus, leads to reward while acting upon the other, the No go stimulus, leads to a timeout") to rapidly increase performance after the switch. To investigate this possibility, we trained mice on an auditory Go/No go discrimination task, which produced similar levels of motivation and performance without prior knowledge of spatial rules related to reward contingency. Mice were then switched to the spatial visual task (Fig. 2a). The stimuli used for the auditory task were two pure tones played from the same speaker, bearing no clear spatial relationship to the visual stimuli introduced after the modality switch. In absence of spatial prior, performance dropped to chance level after the switch but steadily recovered to expert level over the next few sessions (Fig. 2b and Supplementary Fig. 1d, e). We concluded that mice could learn the task after modality switch significantly faster in absence of the conflicting spatial prior observed for mice experiencing rule-reversing modality switches (Fig. 2c).

We further confirmed that mice could also generalize the task rule from the visual to the tactile modality, demonstrating bidirectional cross-modal generalization of learned associations (Fig. 2d–f and Supplementary Fig. 1f–h). Notably we observed a consistent drop in performance during switches from visual to tactile modalities under rule-preserving conditions, an effect that was absent during tactile-to-visual switches. This asymmetry likely reflects differences in task difficulty specific to each modality, leading to distinct performance ceilings for the visual and tactile tasks, regardless of whether they were performed before or after the modality switch (Fig. 2g). The difference in task difficulty may stem from lower discriminability capabilities between nearby whiskers compared to small visual stimuli, despite their matching spatial locations. Consequently, task performance is expected to remain stable or improve slightly when switching from a tactile to a visual task. In contrast, a performance decline is anticipated when switching from a visual to a tactile task, consistent with our observations. None of the observed behavioral effects could be attributed to differences in learning trajectory, expertise level, or motivational state, as the different cohorts of mice were exposed to a comparable number of trials, performed similarly, and exhibited the same weight loss (Supplementary Fig. 2).

Finally, we compared our results for rule-reversing modality switches with conditions in which reward contingencies were switched within the tactile modality, as previously studied[21,22]. After mice reached expert level in the tactile task, we reversed the reward contingencies between the two whiskers. This led to a stark drop in discrimination performance below chance level, indicating that mice rigidly persisted in performing the task following the original rule (Supplementary Fig. 4). Following the switch, performance increased but remained below or at chance level for at least three consecutive sessions. Thus, mice behave differently when reward contingencies are switched within the same sensory modality as they continue to inflexibly produce the same sensorimotor transformation, likely reflecting ingrained habitual behaviors.

## Co-aligned visuo-tactile spatial maps in the dorsal cortex

Our behavioral results suggest that mice can generalize previously learned sensorimotor associations across sensory modalities (Supplementary Fig. 5), based on a common representation of the peripersonal space. Cortical circuits are necessary for perception and are believed to mediate flexible goal-directed behaviors such as cross-modal generalization[2,23]. To pinpoint the cortical regions responsible for visuo-tactile generalization, we mapped the topographic representation of vertical space for both modalities in the dorsal cortex of transgenic mice expressing the calcium indicator GCaMP6f in cortical layers 2/3 (Fig. 3a, see "Methods"). We first used established retinotopic mapping protocols[20,24] to identify whisker-responding and retinotopically organized cortical areas through a 5 mm diameter cranial window over the posterior part of the left dorsal cortex (Supplementary Fig. 6a). Whisker response patterns and retinotopic sign maps enabled us to precisely fit a projection of the Allen

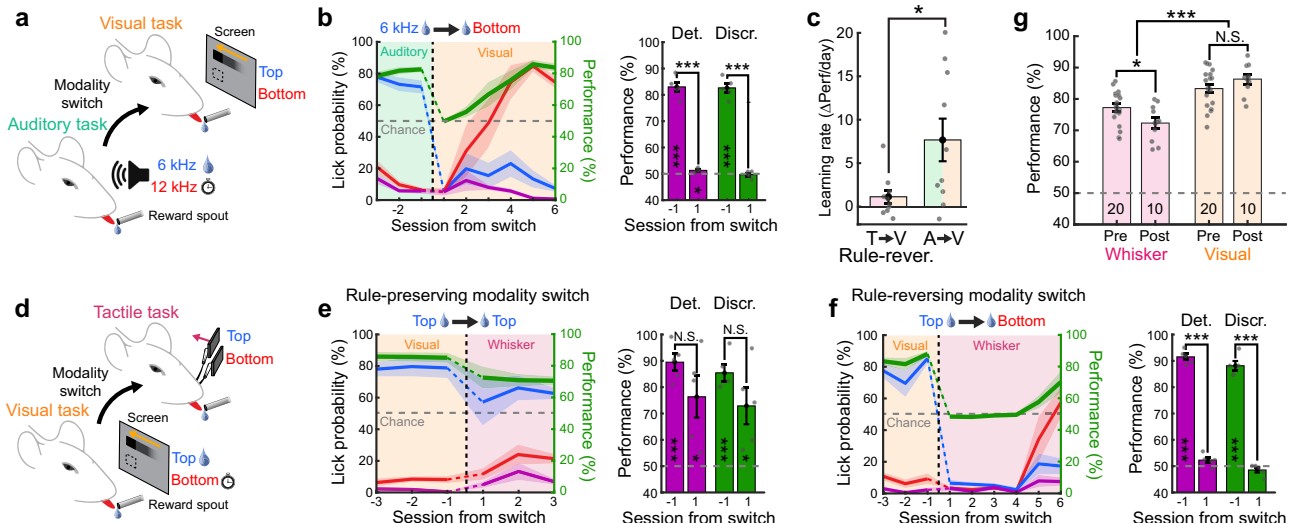

**Fig. 2 | Bidirectional cross-modal generalization of spatial information.**
**a** Schematic of the behavioral paradigm where switches occur between an auditory discrimination task with two pure tones (6 kHz and 12 kHz) and a visual task. The 6 kHz tone only is associated with a water reward. **b** Left: Average task performance and conditional lick probabilities across sessions for mice ($N = 5$ mice) undergoing a modality switch (dashed vertical line). Shaded areas and color code as in Fig. 1e. Right: detection (purple) and discrimination (green) performance distribution for the session before and after the switch (two-sided paired $t$ test comparing days, Det.: ***$p = 1.2 \times 10^{-7}$; Discr.: ***$p = 6.6 \times 10^{-5}$). Performances are also tested against chance level (two-sided $t$ test, Det.: ***$p = 4.8 \times 10^{-5}$ and *$p = 0.031$; Discr.: ***$p = 4.4 \times 10^{-5}$ and Blank $p = 0.59$). Error bars: S.E.M. **c** Comparison of relearning rates between mice that underwent a rule-reversing tacto-visual modality switch and mice that underwent switch from a non-spatial auditory task to the same visual task ($N = 10$ mice for tactile group and $N = 10$ mice for auditory group, unpaired two-sided $t$ test, *$p = 0.02$). Error bars: S.E.M. **d** Schematic of the behavioral

paradigm, where switches occur between a visual task and a tactile task with the top visual stimulus being the rewarded one. **e** Same as panel b for mice undergoing a rule-preserving switch ($N = 5$ mice, two-sided paired $t$ test comparing days, Det.: N.S. $p = 0.16$; Discr.: N.S. $p = 0.29$). Performances are also tested against chance level (two-sided $t$ test, Det.: ***$p = 2.3 \times 10^{-4}$ and *$p = 0.029$; Discr.: ***$p = 3.5 \times 10^{-4}$ and *$p = 0.03$). **f** Same as panel b but for a rule-reversing modality switch ($N = 5$ mice, two-sided paired $t$ test comparing days, Det.: ***$p = 7 \times 10^{-9}$; Discr.: ***$p = 7.2 \times 10^{-5}$). Performances are also tested against chance level (two-sided $t$ test, Det.: ***$p = 3.6 \times 10^{-6}$ and Blank $p = 0.098$; Discr.: ***$p = 2.9 \times 10^{-5}$ and Blank $p = 0.2$). **g** Comparison of average task performance in the tactile and visual tasks before (pre) or after (post) modality switches across mice (sample size indicated in the bar plot). Only mice that underwent rule-preserving switches were included after the switch (two-sided unpaired Wilcoxon test, *$p = 0.04$, ***$p = 2.3 \times 10^{-6}$, N.S. Not significant $p = 0.22$). Error bars: S.E.M.

Mouse Brain Atlas to the cranial window of each mouse (see "Methods"). We used this atlas to register all functional maps into a common reference frame (Supplementary Fig. 6b–f).

To characterize the cortical representation of the vertical space for both unisensory and multisensory stimulations, we designed a visuo-tactile sparse noise protocol (Fig. 3a). Tactile stimuli consisted of single whisker deflections applied to either the top or bottom whisker, while visual stimuli were black squares drifting in the rostro-caudal direction, displayed at eight different vertical locations. Visual and tactile stimuli were presented either individually or simultaneously in all possible combinations (see "Methods"). This protocol was used in task-naïve mice to obtain retinotopic and somatotopic maps for vertical space by computing the preferred spatial position for each pixel. In response to whisker stimuli, we identified the well-established somatotopic arrangement of the primary and secondary whisker somatosensory cortices, S1 and S2, which exhibited a topographic inversion at their boundary (Fig. 3b). Importantly, we observed that whisker stimulations also evoked organized somatotopic maps in several visually responsive areas including the anterior (A), rostro-lateral (RL), antero-lateral (AL) and latero-intermediate (LI) areas. This suggested that whisker representations might be present in a more extended cortical network than previously reported[25]. Strikingly, the maps we obtained with visual stimuli displayed a very similar organization to the tactile ones in these associative areas as well as in S1 and S2 (Fig. 3c). The extended spatial representations evoked by visual or tactile stimuli were found consistently across mice (Supplementary Fig. 6g). This suggests that spatially localized stimuli, regardless of their visual or tactile nature, might share a common topographic representation that facilitates mapping

between sensory modalities, as previously observed in the superior colliculus[16,17]. Spatial representations evoked by these two modalities displayed an angular offset that we estimated around 30 degrees by comparing the angle difference between gradient vectors obtained from these maps (see "Methods"). This might reflect the mouse's internal model of how whisker sensations align with their corresponding locations in the visual field[13].

We further investigated the functional properties of these representations by first computing a modality preference index to assess what sensory modality dominates each area (Fig. 3d). As expected, S1 and S2 were dominated by tactile inputs whereas V1 was dominated by visual inputs. RL and the region at the border between AL and LI displayed a more balanced preference for both modalities. In addition, we measured the spatial coherence between maps of vertical retinotopy and somatotopy indicating local co-alignment (Fig. 3e, see "Methods"). This confirmed a widespread spatial co-alignment across most associative areas in the belt between V1 and S1. We further computed a multisensory modulation index, comparing multisensory responses triggered by visuo-tactile stimuli to the maximal unisensory response on a pixel-by-pixel basis (see "Methods"). The resulting map revealed a strong multisensory enhancement in visuo-tactile associative areas and in S1 (Fig. 3f). Multisensory responses were comparable under visuo-tactile conditions where whisker stimuli were synchronous or delayed by 0.15 s to ensure simultaneity of evoked responses in the cortex (Supplementary Fig. 7), with a tendency to show stronger enhancement in the latter case, as previously documented[19]. Moreover, we found that multisensory enhancement was more pronounced in regions with higher coherence between spatial maps and with strong bimodal representations (Fig. 3g, h).

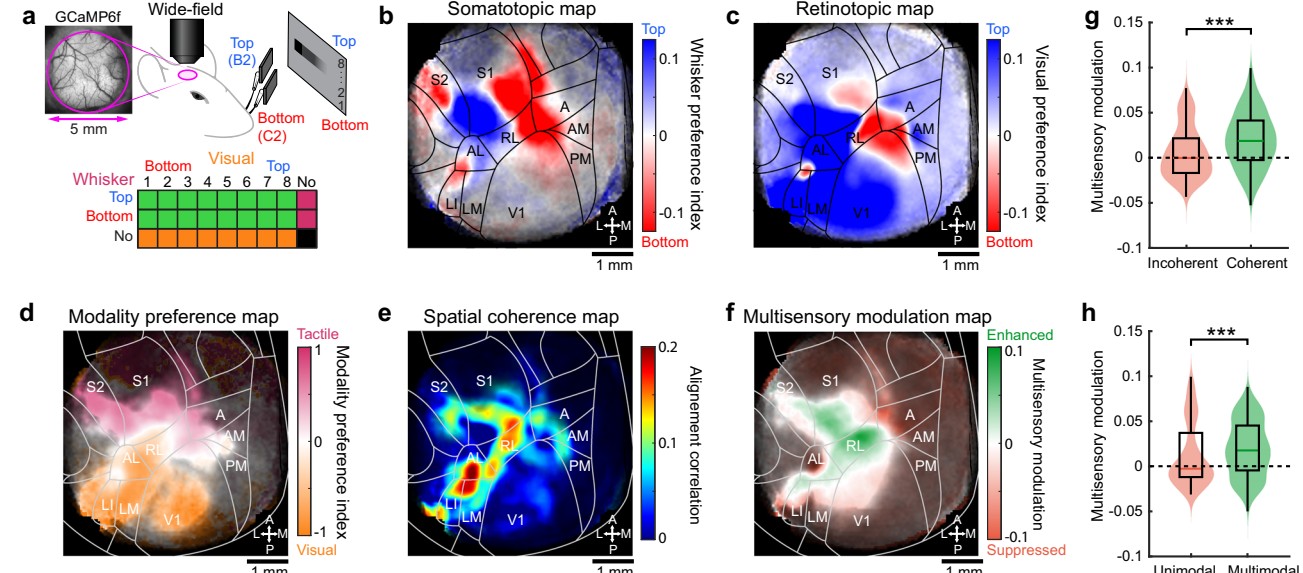

**Fig. 3 | Co-aligned visual and tactile functional maps in the dorsal cortex. a** Top: schematic of the visuo-tactile sparse noise protocol used during wide-field imaging of GCaMP6f-expressing mice through a cranial window. Bottom: matrix of all combinations of visual and whisker stimuli (green: multisensory, orange: visual, magenta: tactile). **b** Somatotopic map of vertical space computed from whisker stimuli averaged across mice ($N = 29$ mice for all panels of the figure), with transparency defined by response significance in each pixel (see "Methods"). A projection of the Allen Mouse Brain Atlas is overlaid on top with areas names and orientation. **c** Average retinotopic map of vertical space. **d** Average modality preference map between visual and tactile responses. **e** Average spatial coherence map

between visual and tactile representations (see "Methods"). **f** Average multisensory modulation map comparing visuo-tactile responses and combination of unisensory responses (see "Methods"). **g** Multisensory modulation index for pixels belonging in regions of high spatial coherence compared to regions without spatial coherence ($n = 26,286$ pixels versus $n = 28,394$ pixels, unpaired two-sided Wilcoxon test, ***$p < 10^{-300}$). **h** Multisensory modulation index for pixels belonging to unimodal or multimodal regions ($n = 10,264$ pixels versus $n = 15,921$ pixels, unpaired two-sided *Wilcoxon*, ***$p < 10^{-116}$). Violin plots show the data distribution (the violin outline), while the overlaid box indicates the median (center line), interquartile range (bounds of the box), and 1.5× interquartile range (whiskers).

## Anatomical origin of spatial maps in the dorsal cortex

Functional maps measured with wide-field calcium imaging could result from direct inputs from visual and tactile primary cortical areas, from evoked top-down inputs[26], or even be the result of highly stereotypical uninstructed movements evoked by sensory stimuli[27]. To investigate the synaptic origin of these maps, we performed anatomical experiments to map both feedforward and feedback projections between primary sensory areas and associative areas displaying visuo-tactile representations (Fig. 4a). We obtained visual and tactile functional maps for representation of vertical space in wild-type mice using intrinsic optical signal imaging under low isoflurane anesthesia (Supplementary Fig. 8a, b, d, e, see "Methods"). These maps were then used to identify two cortical locations representing distinct iso-horizontal vertical positions in V1 or to target B2 and C2 barrels in whisker S1. We then opened the cranial window and injected two adeno-associated viral vectors to induce expression of tracer proteins GFP and tdTomato in the respective locations (Supplementary Fig. 8c, f). After 10–15 days, transcardial perfusions were performed and brains were extracted, flattened, and sliced (see "Methods"). Enriched M2 subtype muscarinic acetylcholine receptors (M2 AChR) in V1 and S1 barrel field were used as landmarks to fit the Allen Mouse Brain Atlas on the reconstructed stack, confirming the location of injection sites along the vertical representation of V1 and S1 (Fig. 4b, c). Axonal projections from primary sensory areas were found in associative cortical regions where visuo-tactile responses were measured with the same spatial organization. Variability in injection sites allowed us to create an anatomical map that described axonal preferences for top and bottom location (Fig. 4b, c), aligning closely with wide-field imaging results (Fig. 3b, c). This confirmed that the functional maps are inherited, at least in part, from direct feedforward projections from primary cortical areas.

While visual stimuli could evoke organized responses in S1 (Fig. 3c), no direct projections were found between V1 and S1[28] suggesting the existence of spatially organized feedback projections from

associative areas to S1. Previous work has shown that feedback projections from higher visual areas (including A, RL, AL and LI) to V1 are spatially organized along the vertical dimension[29] but it is not clear if this holds true for feedback projections to S1. Feedback projections from associative areas to S1 were characterized using the same strategy, but with injections of Cholera Toxin Subunit B (CTB) conjugated either with Alexa 555 or Alexa 647 (Fig. 4d). S1-projecting neurons for top and bottom locations aligned with wide-field imaging (Fig. 3b) and anterograde tracing (Fig. 4b). We additionally confirmed that the same was true for V1 (Supplementary Fig. 8g) as previously reported[29]. Thus, we observed a shared spatial organization of feedforward and feedback projections between primary and associative areas. Using retrograde labeling with CTB injections in RL, we also compared the cell density of RL-projecting neurons in V1 and S1, revealing that V1 contains a denser population of neurons projecting to RL (Supplementary Fig. 8h–k). This asymmetry, together with feedback connections from associative areas to primary cortices, could explain why visual stimuli evoked stronger responses in S1 than the other way around (Fig. 3b, c).

## Single-neuron visuo-tactile functional properties

Spatially organized feedforward and feedback projections could facilitate generalized sensorimotor learning through transfer of spatial information across sensory modalities. However, functional maps obtained with wide-field imaging do not reveal precise computations performed at single-cell level and could still be prone to artifacts produced by neuronal processes originating from other brain structures. To overcome this limitation and assess if neurons in associative areas can mediate cross-modal generalization, we performed two-photon calcium imaging in a subset of mice (Fig. 5a). Single neuron somatic GCaMP6f signal was extracted during the visuo-tactile sparse noise protocol in fields-of-view covering different cortical areas identified with the atlas (Supplementary Fig. 9a–f). Given the observed fluorescence response patterns to various unimodal and multimodal

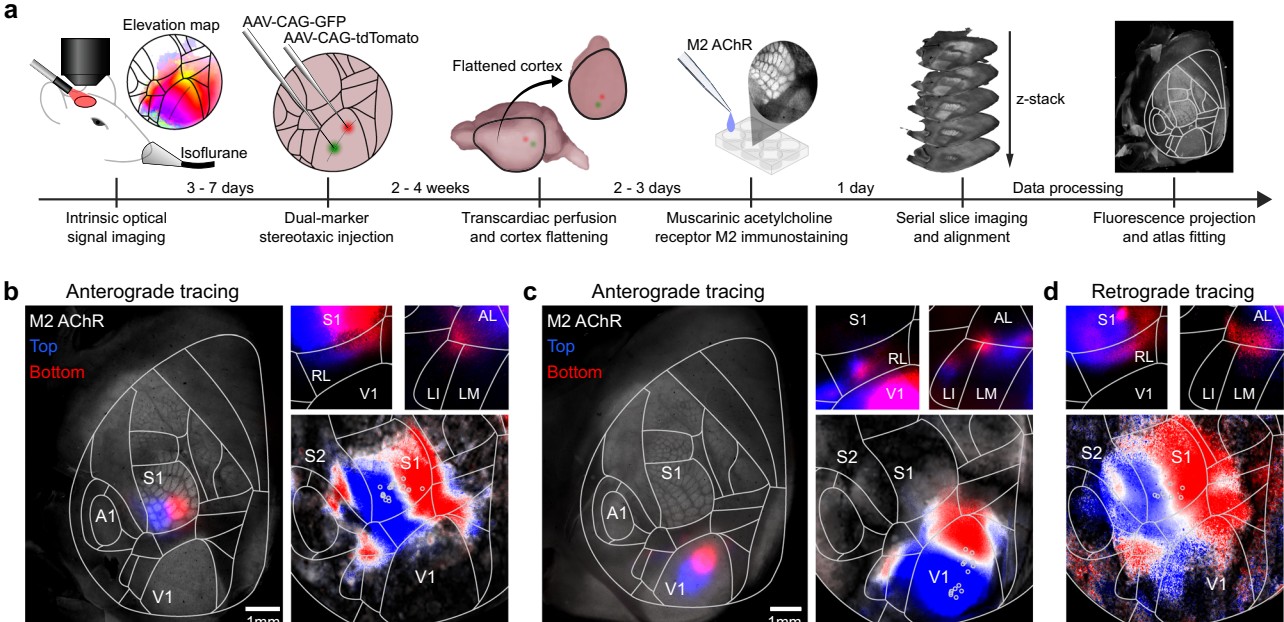

**Fig. 4 | Co-aligned visual and tactile anatomical projection maps in the dorsal cortex. a** Schematic of the data collection and analysis pipeline for topographic anatomical tracing. **b** Anterograde labeling of S1 projections using injections of viral vectors AAV-CAG-GFP and AAV-CAG-tdTomato. Left: injection sites in B2 and C2 barrels in whisker S1. Right: conserved somatotopic organization of projections in associative areas for the cortex shown on the left (top). Bottom: Reconstructed whisker preference averaged across mice using all injection sites depicted by circles (see "Methods", $N = 9$ mice and $n = 18$ injections, Pearson coefficient of correlation between anatomical and wide-field map: 0.8311, two-sided $t$ test $p < 10^{-300}$). **c** Same

as panel b but for anterograde labeling of V1 projections with two injection sites along the iso-horizontal axis ($N = 7$ mice and $n = 14$ injections, Pearson coefficient of correlation: 0.5121, two-sided $t$ test $p < 10^{-300}$). **d** Retrograde labeling of S1-projecting neurons using CTB-Alexa 555 and CTB-Alexa 647 injections. Top: Examples of CTB-labeled neurons spatially organized in associative areas. Bottom: Reconstructed map of preferred whisker in projecting neurons over the dorsal cortex combining all injection sites ($N = 5$ mice and $n = 8$ injections, Pearson coefficient of correlation: 0.663, two-sided $t$ test $p < 10^{-300}$).

stimuli (Fig. 5b), we could map specific functional properties of single neurons onto the reference atlas, pooling data across mice, for comparison with corresponding wide-field regions. Many recordings were performed across a large portion of the dorsal cortex to extensively cover responsive visuo-tactile areas (Supplementary Fig. 9g, h). We reconstructed the somatotopic map of vertical space across the cranial window (Fig. 5c), which closely aligned with the wide-field map (Fig. 3b). Neurons with whisker tactile responses were found across the belt of associative areas following the somatotopic organization. This further confirmed the existence of an extended network of whisker responsive and visuo-tactile cortical regions[25]. The same was true for neurons responding to visual stimuli, which were found across most visual and tactile areas, including S1 (Fig. 5d), consistent with the responses observed using wide-field imaging (Fig. 3c). Reconstructed population maps based on single-neuron properties consistently matched with functional maps measured with wide-field calcium imaging (Supplementary Fig. 10).

Among all neurons imaged, we identified four functional cell-types. Unisensory visual or tactile neurons only responded to their respective modality, gated neurons responded only when both visual and tactile stimuli were presented together, and bimodal visuo-tactile neurons responded to both unisensory visual and whisker stimuli (Supplementary Fig. 11a, b). Importantly, bimodal visuo-tactile neurons localized into two distinct clusters (Fig. 5e), corresponding to the multimodal domains identified by wide-field imaging (Fig. 3d), and consistent with anatomical projection patterns (Fig. 4b, c). As these clusters coincided with areas associated with the dorsal (A, RL) and ventral (AL, LI) streams, we used these designations moving forward. Due to their distinct functional properties, visuo-tactile neurons may facilitate cross-modal generalization. A Bayesian decoder trained to discriminate whisker stimuli based on individual neuronal responses was evaluated for generalization to visual stimuli (see "Methods").

Visuo-tactile neurons in associative areas uniquely enabled effective generalization across sensory modalities (see Fig. 5f). Other neuron types demonstrated significantly lower decoding accuracy when we tested their ability to generalize the tactile discrimination to the visual modality, particularly with larger populations (see Supplementary Fig. 11c–e). Therefore, visuo-tactile neuronal population, displaying a prominent preference for spatially congruent combinations in task-naïve mice, possesses the necessary properties to mediate goal-directed cross-modal generalization if decoded by a downstream decision-related brain area[30]. This suggests that cross-modal generalization might occur without the need for task-induced synaptic plasticity in sensory circuits.

We further characterized the response properties of single neurons for visuo-tactile stimuli in comparison with their responses predicted from unisensory responses (see "Methods"). We found that bimodal neurons were typically tuned to spatially congruent stimuli across both modalities. The example neuron in Fig. 5b responded preferentially to the bottom part of the visual field and to the bottom whisker. Using unisensory responses, we predicted the response pattern to visuo-tactile stimuli as the maximum response between the two modalities for each combination. When comparing the predicted response with the measured one, we observed suppression in incongruent combinations and enhancement in congruent ones. This non-linear modulation profile enables neurons to maintain their spatial selectivity independently of the stimulated modality (visual, tactile, or visuo-tactile). This response property is reminiscent of the supramodal encoding of object orientation reported in visuo-tactile neurons of the rat posterior parietal cortex[6]. Population analysis confirmed that spatial congruence in multisensory selectivity is prevalent among populations of multimodal neurons in both the ventral and dorsal stream (Fig. 5g). Additionally, the two neuronal populations exhibited sharper tuning for whisker and visual positions than that predicted by their

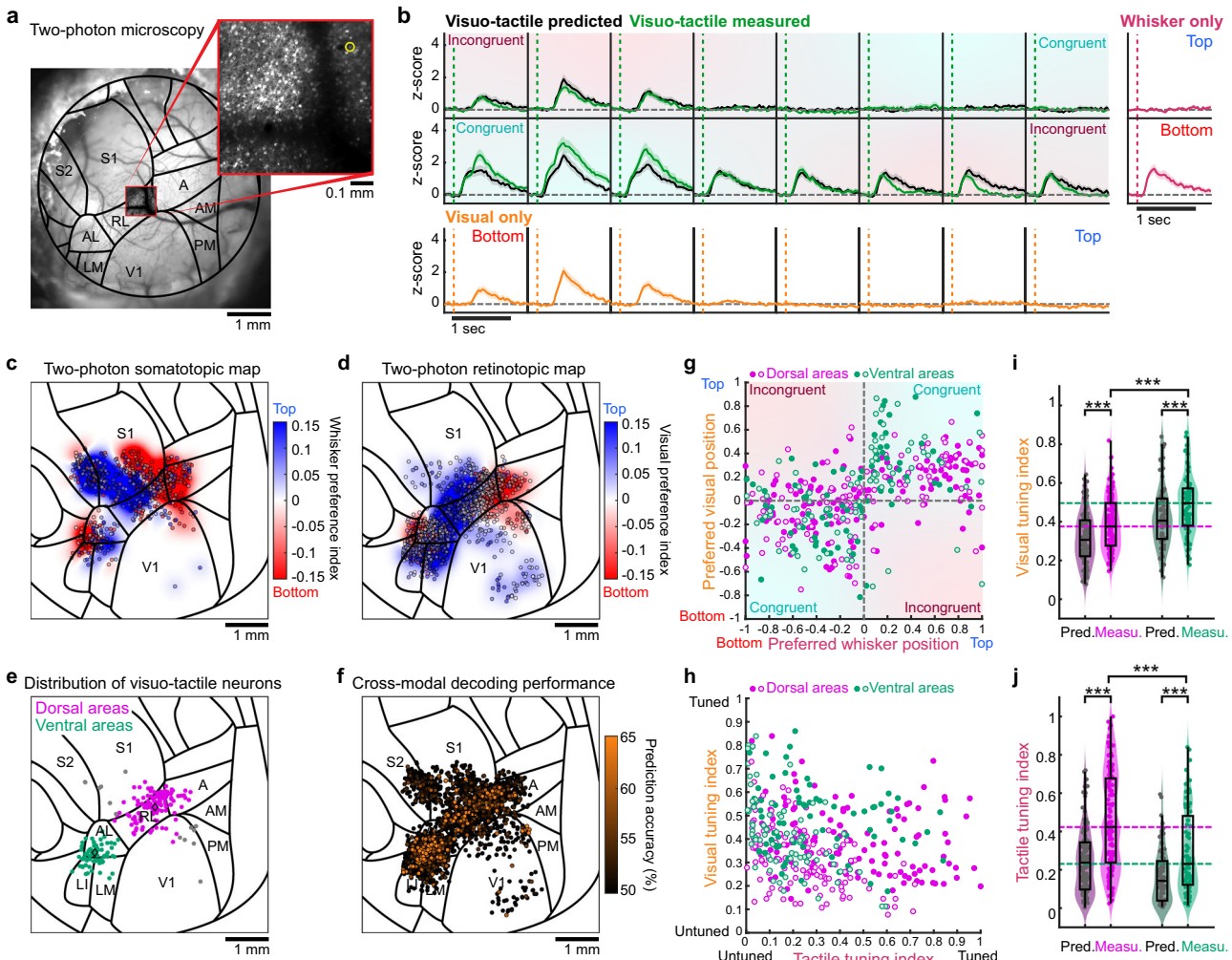

**Fig. 5 | Visuo-tactile representation of peri-personal space in single neurons.**
**a** Cranial window and two-photon field-of-view overlapping with area RL.
**b** Response pattern of GCaMP6f for the neuron highlighted in panel a (yellow circle). Unisensory z-scored responses for visual (orange) or whisker (magenta) stimuli. Predicted multisensory responses are shown for visuo-tactile stimuli (black) together with measured responses (green). Shaded areas: S.E.M. **c** Single neurons with significant responses to whisker stimuli ($n = 567$ neurons, $N = 25$ mice). Color code indicates preference for top (blue) or bottom (red) stimuli. Reconstructed wide-field map in the background (see "Methods", Pearson coefficient of correlation: 0.8472, two-sided $t$ test $p < 10^{-300}$). **d** Same as panel c for neurons significantly responding to visual stimuli ($n = 1593$ neurons, Pearson coefficient of correlation: 0.6505, two-sided $t$ test $p < 10^{-300}$). **e** Distribution of all visuo-tactile bimodal neurons. Neurons are classified as part of the ventral (green) or dorsal (pink) pathway depending on their location (see "Methods"). Neurons outside these pathways in gray. **f** Distribution of responsive neurons color-coded by their visual decoding accuracy, following training with whisker-stimulation responses (see "Methods", $N = 2563$ neurons, $N = 25$ mice). **g** Comparison of preferred visual position and preferred whisker in visuo-tactile stimulations condition for predicted (open circles) and measured (full circles) responses of ventral neurons ($n = 75$ significantly responsive multimodal neurons, Pearson coefficient of

correlation: 0.36 for predicted with two-sided $t$ test $p = 1.6 \times 10^{-3}$ and 0.48 for measured with two-sided $t$ test $p = 1.5 \times 10^{-5}$; 77% of neurons in congruent quadrants) and dorsal neurons ($n = 124$ significantly responsive multimodal neurons, Pearson coefficient of correlation: 0.55 for predicted with two-sided $t$ test $p = 3.7 \times 10^{-11}$ and 0.47 for measured with two-sided $t$ test $p = 4.3 \times 10^{-8}$; 71% of neurons in congruent quadrants). **h** Comparison between tactile and visual tuning indices computed from the predicted (open circles) or measured (full circles) visuo-tactile responses of neurons in panel e. **i** Comparison of the visual tuning indices from panel h between predicted and measured responses for ventral and dorsal stream neurons (two-sided paired Wilcoxon test between measured and predicted for ventral: $n = 75$ neurons, ***$p = 1.7 \times 10^{-5}$; for dorsal: $n = 124$ neurons, ***$p = 3.1 \times 10^{-11}$; two-sided unpaired $t$ test comparing dorsal and ventral measured responses: ***$p = 3.2 \times 10^{-5}$). **j** Same as panel i for the tactile tuning indices (two-sided paired Wilcoxon test between measured and predicted for ventral: $n = 75$ neurons, ***$p = 7.7 \times 10^{-10}$; for dorsal: $n = 124$ neurons, ***$p = 1.6 \times 10^{-19}$; two-sided unpaired $t$ test comparing dorsal and ventral measured responses: ***$p = 1.1 \times 10^{-5}$). Violin plots show the data distribution (the violin outline), while the overlaid box indicates the median (center line), interquartile range (bounds of the box), and 1.5× interquartile range (whiskers).

unisensory responses alone (Fig. 5h). Visuo-tactile neurons were found in most mice in which imaging was performed in associative areas (18 mice out of 22 for dorsal areas, 14 mice out of 17 for ventral areas). Strong multisensory modulations, with a norm of the difference between observed and predicted selectivity larger than 0.3 (Fig. 5h), were observed in a subset of these neurons (32 neurons out of 124 found in 8 mice out of 18 for dorsal areas, 12 neurons out of 75 found in 6 mice out of 14 for ventral areas). These modulations were

significantly stronger in neurons from dorsal areas compared to those from ventral areas ($n = 75$ neurons for ventral versus $n = 124$ neurons for dorsal, unpaired two-sided $t$ test, ***$p = 7.6 \times 10^{-4}$ with average norm difference $= 0.203 \pm 0.015$ for ventral and average norm difference $= 0.122 \pm 0.018$ for dorsal). In particular, neurons from the ventral domain displayed sharper tuning for particular visual locations, whereas neurons in the dorsal domain exhibited greater tuning to individual whiskers (Fig. 5i, j), consistent with modality preferences

observed in wide-field imaging data (Fig. 3d). These results showed that bimodal neurons in visuo-tactile associative areas are indeed specifically responding to spatially congruent visuo-tactile stimuli and therefore could mediate cross-modal generalization during goal-directed tasks.

## Cross-modal generalization loss-of-function

Cross-modal decodability of stimulus location observed at single-cell level (Fig. 5f), alongside the anatomical connectivity between visual and whisker somatosensory cortices (Fig. 4), suggest that associative visuo-tactile cortical areas could mediate cross-modal generalization of goal-directed behaviors. We performed loss-of-function experiments to assess the necessity of these areas for visuo-tactile generalization. Given that task-naïve mice (those not yet exposed to any behavioral tasks) exhibited co-aligned and anatomically connected spatial maps, we reasoned that the reverberation of evoked responses across this extensive visuo-tactile network could potentially drive learning processes beyond the initially stimulated modality. This could facilitate cross-modal generalization before any direct exposure to the second task with the other modality. Indeed, the existence of supra-modal representations of space in associative areas could shape sensorimotor circuits across sensory modalities that share the same spatial properties during the learning of the first task. To prevent this possibility, we decided to chronically silence associative cortical areas prior to any sensorimotor learning. We virally expressed the tetanus toxin light chain (TeNT-P2A-GFP) in ventral or dorsal areas, thereby preventing neurotransmitter vesicle release in transfected neurons (Fig. 6a, see "Methods"). Neurons expressing TeNT also co-expressed GFP, allowing comparison of the expression pattern with the fitted atlas (Fig. 6b). After subtracting blood vessel patterns and comparing with the cranial window before injection (see "Methods"), we characterized the extent of GFP expression and overlap with different visuo-tactile areas (Supplementary Fig. 12a–d). To ensure strong expression of TeNT to effectively suppress vesicle release in the transfected neurons before the beginning of the behavioral training, we waited at least four weeks after viral injections[31]. Mice learned the whisker discrimination task at the same rate as control mice and reached expert performances comparable to those observed in mice trained without TeNT expression (Fig. 6c, Supplementary Fig. 12a–d). Additionally, mice expressing TeNT performed a comparable number of trials to control mice ($N = 20$ mice for control group, $N = 25$ mice for TeNT group, unpaired two-sided $t$ test, $p = 0.89$). However, when switching to the visual task in the rule-preserving condition, performance typically dropped to chance level (Fig. 6d, k).

To identify which visuo-tactile areas are necessary for cross-modal generalization, we expressed TeNT-P2A-GFP in different cortical locations across mice (see "Methods"). We were able to cover all visuo-tactile areas with varying degrees of overlap with the expression of TeNT (Supplementary Fig. 12e). Injections performed in ventral or dorsal associative areas resulted in comparable expression patterns at the surface of the cortex after correcting for any obstruction caused at the edge of the cranial window (Supplementary Fig. 12f). The extent of TeNT-P2A-GFP expression varied across mice and could overlap with multiple areas, requiring correlative analyses that accounted for these varying degrees of overlap or were independent of specific areas. Taking advantage of this variability, we first calculated for each cortical area the extent to which the overlap of its surface with TeNT expression correlated with any decline in performance following a rule-preserving switch (Supplementary Fig. 12g). This analysis revealed that generalization impairment was significantly correlated with TeNT expression in area RL only (Fig. 6e). Despite the proximity of RL to S1 and V1 and the risk of inactivating these primary sensory cortices with TeNT, we did not observe any significant correlation in these areas (S1 and V1 in Fig. 6e). To confirm this result, we also performed an area-independent reverse correlation analysis mapping the average TeNT

coverage that evoked a complete impairment of cross-modal generalization (see "Methods"). Here again, RL silencing was found to consistently prevent cross-modal generalization, revealing that this associative area is necessary for the transfer to occur (Fig. 6f). Grouping mice depending on whether TeNT expression pattern was located in the dorsal or ventral stream (Fig. 6g, i), we confirmed that dorsal stream silencing severely affected generalization performance (Fig. 6h), leading to slower re-learning after rule-preserving switches. Although mice expressing TeNT in RL initially failed to detect or discriminate visual stimuli after the task switch, they regained performance in the visual task after several training sessions. This indicates that despite a comparable capacity to perform the visual or tactile task, these mice were not able to generalize learning across sensory modalities.

In contrast, inactivation of ventral areas slightly impaired performance immediately after a rule-preserving switch, although mice were performing on average well above chance level (Fig. 6j). A direct performance comparison between mice injected in dorsal versus ventral areas during the four days following the modality switch confirmed a significant difference between these cohorts ($N = 8$ mice for dorsal, $n = 7$ mice for ventral; discrimination performance: unpaired two-sided $t$ test, $p = 0.036$; detection performance: unpaired two-sided $t$ test, $p = 0.046$). These findings support the conclusion that silencing RL impairs cross-modal generalization without excluding the possibility that ventral regions such as AL may play a minor part, particularly given the transient performance drop observed in these mice. One potential confounding factor is the imprecision of estimating TeNT-P2A-GFP expression exclusively from surface observations. The spread of expression within deeper layers, undetectable from the surface, could occasionally occur and affect nearby areas such as RL without being identified through wide-field imaging. These results suggest a predominant effect of RL silencing in impairing cross-modal generalization abilities but not unisensory skills.

We further assessed how silencing area RL with TeNT affects task performance following rule-reversing modality switch (Supplementary Fig. 13a, b). Following the switch, their performance fell to chance level but recovered faster than what we observed for control mice (Supplementary Fig. 13c). To fully characterize the learning rate following modality switches in TeNT-expressing mice, we focused on all mice that exhibited complete generalization impairment (i.e. displaying performance near chance level on the first session following the switch regardless of whether the condition was rule-reversing or rule-preserving). For these mice, we quantified the re-learning rate over several sessions following the modality switch and observed that the distribution was comparable to that seen during non-spatial auditory-to-visual switches (Fig. 6l). This implies that silencing area RL prevents mice from applying prior knowledge about the spatial task rule, whether it is conflicting or congruent with the new task rule, resulting in the mice learning the visual task without prior spatial knowledge.

## Optogenetic sensory substitution for cross-modal generalization

To determine whether cortical area RL is not only necessary, but also sufficient for visuo-tactile generalization, we conducted gain-of-function experiments by substituting visual stimuli after modality switch with direct optogenetic stimulations of RL top- and bottom-encoding subregions (Fig. 7a). Projections from primary sensory areas to the ventral and dorsal visuo-tactile stream being largely distinct (Supplementary Fig. 8h–j), we assumed that optogenetic activations of RL would remain restricted mostly to the dorsal stream. Functional mapping was first performed to identify domains of RL encoding top and bottom visual stimuli in Ai32 transgenic mice expressing Cre-dependent ChR2-eYFP. Viral vectors AAV1.CaMKIIa.Cre were then injected in this area to broadly express ChR2-eYFP in RL (Fig. 7b). We employed a projector-based microscope to precisely shape blue light

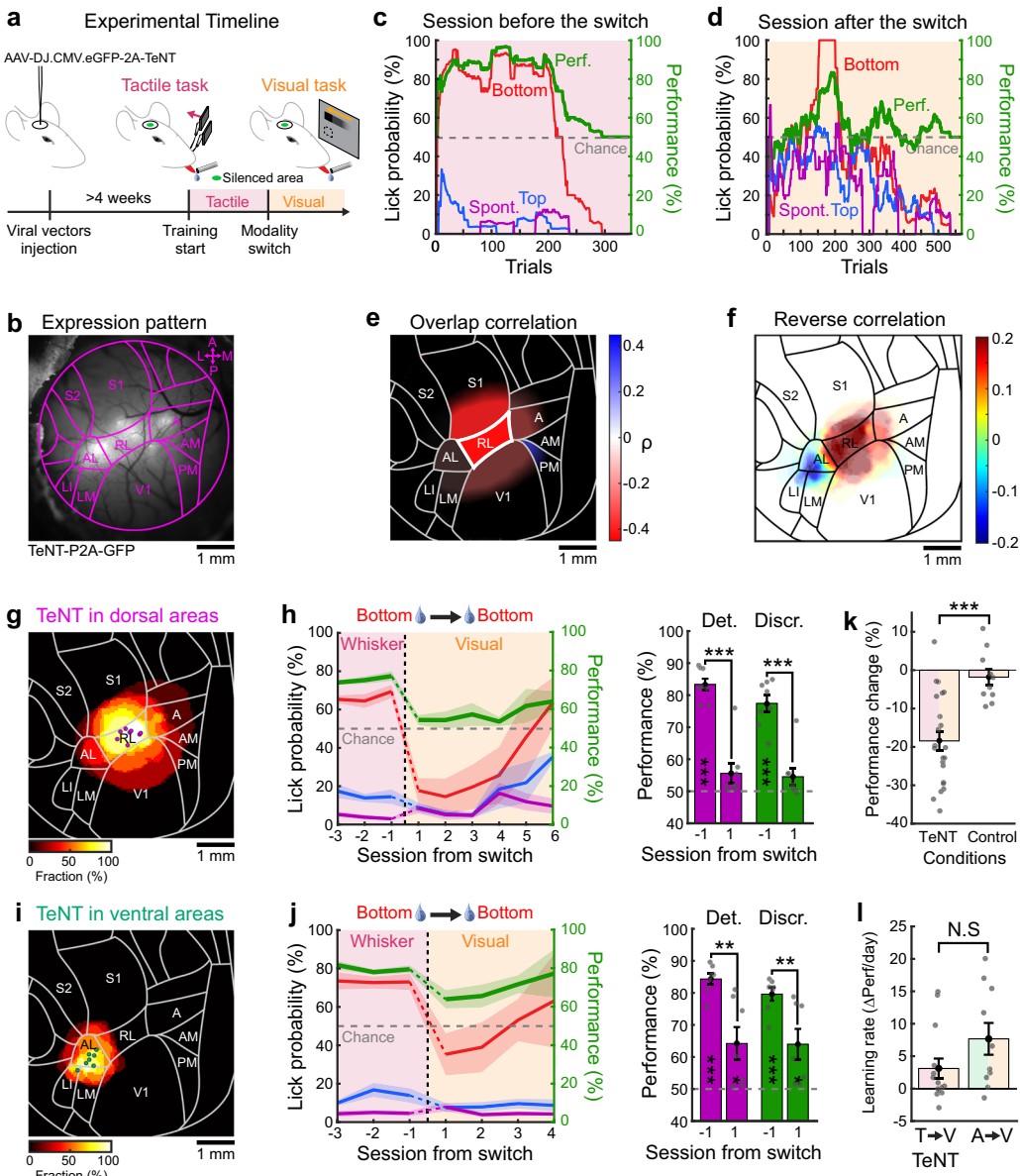

**Fig. 6 | Area RL is necessary for cross-modal generalization. a** Schematic timeline for loss-of-function experiments. **b** Cranial window with TeNT-P2A-GFP expression and atlas overlaid. **c** Example session before a rule-preserving modality switch from a tactile to a visual task. Color code as in Fig. 1d. **d** Session following the modality switch. **e** Area-based correlation between TeNT-P2A-GFP expression overlap and performance drop following modality switch. Color map indicates Pearson coefficients of correlation ρ. Areas with $p < 0.05$ are indicated with a thick border (RL: two-sided $t$ test $p = 0.047$). **f** Average TeNT-P2A-GFP coverage for mice with impaired cross-modal generalization (see "Methods"). The map is displayed after subtraction of the average coverage across all injected mice. **g** Average TeNT-P2A-GFP coverage of mice where only dorsal neurons were silenced ($N = 8$ mice). Dots indicate center-of-mass location for each mouse. **h** Left: Average task performance and conditional lick probabilities across sessions for mice in panel g with rule-preserving modality switch (vertical dashed line). Shaded area: S.E.M. Color code as in panel c. Right: detection (purple) and discrimination (green) performance distribution for the session before and after the switch (two-sided paired $t$ test

comparing days, Det.: ***$p = 1.6 \times 10^{-6}$; Discr.: ***$p = 7.2 \times 10^{-5}$). Performances are also tested against chance level (two-sided $t$ test, Det.: ***$p = 2.9 \times 10^{-7}$ and Blank $p = 0.11$; Discr.: ***$p = 1.2 \times 10^{-5}$ and Blank $p = 0.13$). Error bars: S.E.M. **i** Same as panel g but for TeNT-P2A-GFP expression in the ventral stream ($N = 7$ mice). **j** Same as panel h for ventral areas (two-sided paired $t$ test comparing days, Det.: **$p = 0.003$; Discr.: **$p = 0.008$). Performances are also tested against chance level (two-sided $t$ test, Det.: ***$p = 1.3 \times 10^{-6}$ and *$p = 0.033$; Discr.: ***$p = 6.7 \times 10^{-6}$ and *$p = 0.027$). **k** Comparison of performance change following rule-preserving switch between all mice expressing TeNT-P2A-GFP and control mice described in Fig. 1e and Supplementary Fig. 1b ($N = 22$ mice for TeNT and $N = 10$ mice for control, two-sided unpaired $t$ test, ***$p = 2.6 \times 10^{-4}$). Error bars: S.E.M. **l** Learning rate estimated over first three sessions following modality switch for mice expressing TeNT-P2A-GFP with impaired cross-modal generalization and control mice trained to the auditory task first described in Fig. 2a ($N = 14$ mice for TeNT and $N = 10$ mice for control, two-sided unpaired $t$ test, N.S. $p = 0.11$). Error bars: S.E.M.

patterns, selectively exciting subregions of RL that encode top and bottom stimuli. This approach allowed us to substitute visual stimuli in the upper or lower visual field with precise spatio-temporal optostimulations following the modality switch (see "Methods"). Based on the properties and location of single neurons measured through two-photon imaging (Fig. 5f), we estimated that approximately 65% of RL

visuo-tactile neurons with generalizing properties were activated by these blue light patterns, with roughly 40% located in the bottom region and 25% in the top region.

Before switching to the full optogenetic substitution task, mice that had achieved stable expert performance in the tactile task first underwent optogenetic habituation sessions (Fig. 7a). During these

sessions, they were exposed exclusively to the Go light pattern, with a distribution of 80% Go trials and 20% Catch trials with blue light stimulations outside the cranial window. Once mice exhibited a consistent licking response to optogenetic stimulations, they were transitioned to the full task, which included No go trials. This step was necessary, as stimulations of higher-order sensory areas may induce unfamiliar modulations of perceptual experience rather than generating new percepts[32–34], making them more difficult to be detected. We assessed the ability of mice to generalize previously learned sensorimotor associations from the whisker task to the optogenetic task, both when the spatial rule was preserved or reversed. Following rule-preserving modality switch, mice maintained a comparable level of discrimination performance once No go trials were introduced (Fig. 7c). During the opto-habituation phase, these mice quickly learned to respond to optogenetic stimulations (Fig. 7d). In contrast, mice undergoing the rule-reversing modality switch performed at chance level after the No go stimuli were introduced and gradually relearned the task in the following sessions (Fig. 7e). Additionally, they were significantly slower to respond to optogenetic stimulation during the habituation phase (Fig. 7f), taking roughly twice as long to reach the same level of responsiveness (Fig. 7g). This is most likely due to the spatial prior inherited from the tactile task, which associated the top stimuli with a time-out punishment. This suggests that activity in RL alone is sufficient to induce cross-modal generalization, once mice habituated to optogenetic stimulations. To further confirm that this function was specific to RL, we performed the same experiment targeting area AL of the ventral stream (Fig. 7h), which also contains visuo-tactile neurons (Fig. 5e). Optogenetic activation of AL failed to evoke behavioral responses for at least 7 consecutive days, even when stronger blue light intensities were used or when larger light patterns matching the size of those used over RL were applied (Fig. 7i).

## Network model for generalized sensorimotor learning

Based on our experimental findings on the functional and anatomical organization of visuo-tactile cortical circuits, as well as the effect of RL silencing on cross-modal generalization, we developed a network model to recapitulate the behavior we observed in mice. Our model included areas V1, S1, RL and AL, each comprising two recurrent networks of excitatory and inhibitory neurons encoding "top" or "bottom" spatial location, respectively (Fig. 8a). Although areas RL and AL exhibit comparable functional properties, their cortical connectivity differs markedly. Estimations of cortico-cortical connectivity reported by Wang and colleagues[35] indicate that RL shows stronger feedback projections to S1/V1 and robustly connects to the premotor cortex (M2), which is critical for sensorimotor transformation[20]. In contrast, AL has limited feedback projections to primary sensory areas and very weak connectivity to M2. Based on these observations, in the model only areas S1, V1 and RL projected to reward-modulated decision variables representing a downstream motor area such as M2. Additionally, feedback projections from AL to V1 and S1 were modeled as much weaker than those from RL. Training the model on the whisker discrimination task involved external inputs to S1 (representing thalamic inputs) and reward-modulated learning via Hebbian plasticity of the sensorimotor connections (see "Methods"). In line with our experimental observations, we modeled the sensory cortices such that populations encoding top or bottom stimuli could communicate bidirectionally with populations in associative areas encoding the same spatial locations. This model architecture produces visuo-tactile neuronal populations in RL and AL that were tuned exclusively to spatially congruent visuo-tactile stimuli. Synaptic connections between sensory cortices were not subjected to plasticity, as only the synaptic plasticity of sensorimotor connections was necessary to learn the association between the Go stimulus and the corresponding action following a reward. Model simulations showed that after switching inputs from S1 to V1,

mimicking a rule-preserving switch from the tactile to the visual modality, performance remained stable (Fig. 8b). However, if the spatial rule of the task was reversed upon modality switch, performance dropped below chance level and then slowly recovered (Fig. 8c). This delay in performance recovery was due to the model weights needing to override the previously learned structure before learning the new association. Mimicking the loss-of-function experiments, we silenced the area RL in the model during training, causing the neural network to learn the visual task from scratch after the rule-preserving switch without any spatial prior in the weight structure (Fig. 8d). The absence of strong connections between the primary sensory areas prevented ongoing learning in the non-stimulated area. The model suggests that the activity reverberation mediated by bidirectional connections with area RL induces a consistent weight structure across connected areas. These weights, however, needed to be reorganized following a rule-reversing modality switch, leading to delayed relearning time (Supplementary Fig. 14a–d).

To quantify the effect of different task conditions in the learning trajectory following modality switch in the model, we computed the time needed in simulation steps to recover expert level. Following a rule-preserving modality switch, the time to relearn was near-instantaneous whereas it was much longer following rule-reversing switches (Fig. 8e). Interestingly, when RL was silenced, the time to relearn was in-between these two distributions reflecting that the model learned the task from scratch (Fig. 8e). Silencing area AL, however, had no impact on task generalization. This result remained consistent as long as AL feedback strength was very weak compared to RL (Supplementary Fig. 14e). Although we primarily attributed the performance differences between tactile-to-visual and visual-to-tactile switches to stimulus discriminability (Fig. 2g), we also examined whether asymmetrical connectivity between V1/S1 and RL might play a role in the model. Introducing data-based asymmetry values into our model (Supplementary Fig. 8k) had no impact on rule-preserving switches, where generalization remained near-instantaneous in both directions (Supplementary Fig. 14f). However, in rule-reversing conditions, stronger connections from V1 to RL slowed relearning during visual-to-tactile transitions compared to tactile-to-visual relearning. These findings suggest that, although anatomical asymmetries can affect relearning in rule-reversing scenarios, they are unlikely to explain the performance gap in rule-preserving conditions, which is more plausibly driven by modality-dependent stimulus discriminability. Altogether, this network model recapitulates cross-modal generalization and offers mechanistic insights into the circuit organization and synaptic plasticity required for sensorimotor learning, providing testable hypotheses for further experimental validation. Finally, we compared these results to the time needed to relearn the task after modality switch in all conditions used with our mice (Fig. 8f). In line with the model, mice rapidly recovered high-performance levels after a rule-preserving modality switch. In contrast, time to relearn the task was dramatically prolonged following rule-reversing switches. This was true for both visual stimuli and optostimulations following the switch. In TeNT-expressing mice, however, the time to relearn the task was moderate and similar for both rule-preserving and rule-reversing modality switches. Importantly, the time to relearn in these mice was comparable to the time needed to reach expert level in the auditory task in absence of any spatial prior, reminiscent of the "RL silenced" case in the model. The distribution of peak performance observed for our mice was mostly determined by the nature of the sensory inputs after the switch rather than the switch type (Supplementary Fig. 15).

## Discussion

Our results elucidate the cortical architecture underlying the abstract representation of peri-personal space in mice. We identified the

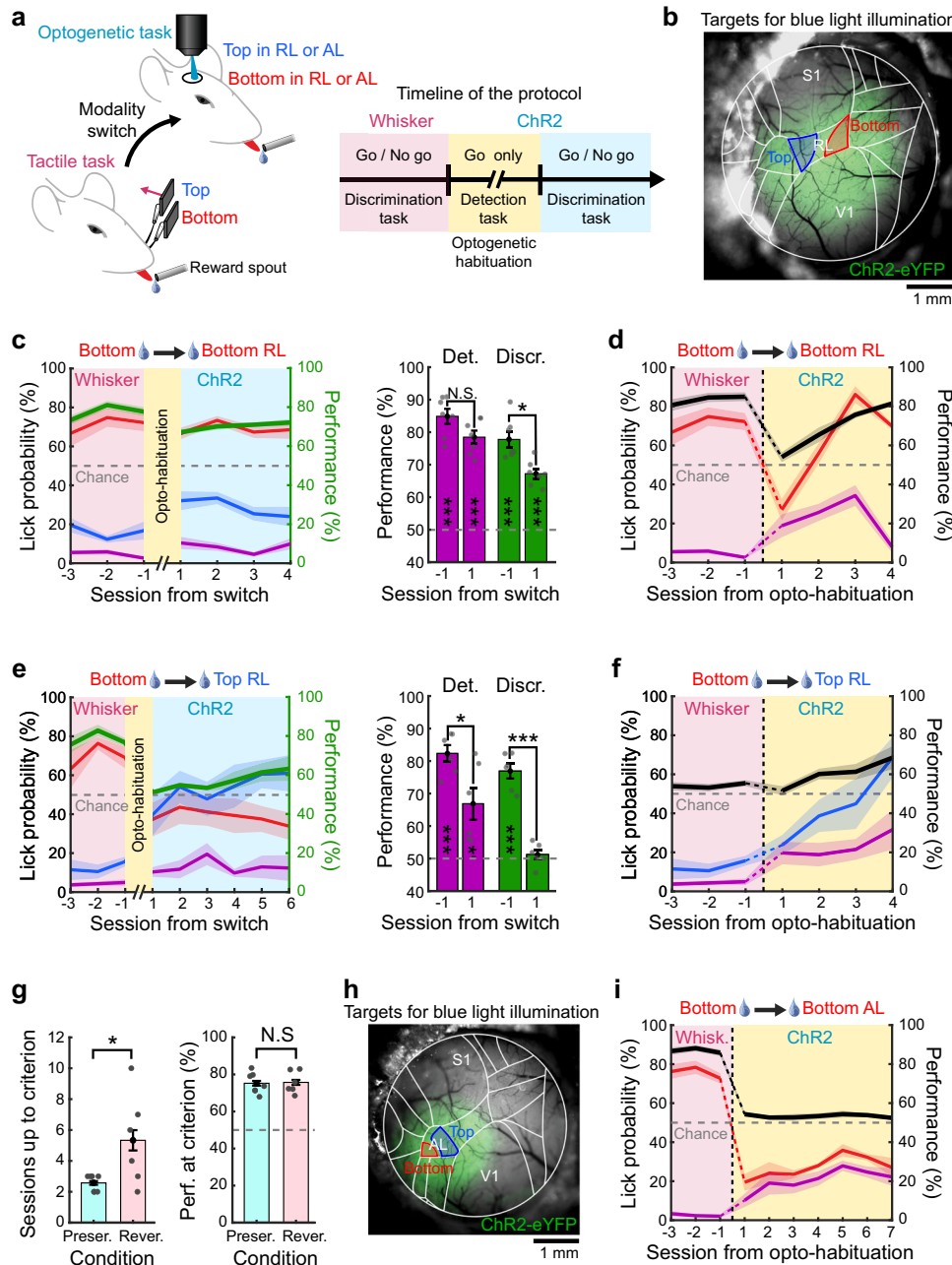

**Fig. 7 | Area RL is sufficient for cross-modal generalization. a** Schematic of the behavioral experiment with direct ChR2-mediated optostimulations of RL sub-regions following modality switch. **b** Example cranial window of an Ai32 mouse expressing ChR2-eYFP after injection of AAV1.CaMKIIa.Cre viral vectors in area RL. Atlas overlaid for reference. Subregions of RL encoding top or bottom stimuli are indicated in blue or red, respectively. Blue light patterns are shaped to match these subregions (see "Methods"). **c** Left: Average task performance and conditional lick probabilities across sessions for mice experiencing a rule-preserving modality switch from the whisker task to the optogenetic task ($N = 7$ mice). Right: detection (purple) and discrimination (green) performance distribution for the session before and after the switch (two-sided paired $t$ test comparing days, Det.: N.S. $p = 0.057$; Discr.: *$p = 0.014$. Performances are also tested against chance level (two-sided $t$ test, Det.: ***$p = 5.1 \times 10^{-6}$ and ***$p = 7.7 \times 10^{-6}$; Discr.: ***$p = 2.3 \times 10^{-5}$ and ***$p = 2.1 \times 10^{-5}$). Error bars: S.E.M. **d** Average detection performance (black) and conditional lick probability for the bottom stimulus (red) or in the absence of stimuli (purple) across sessions for mice switching from the whisker task to the

habituation phase to respond to optogenetic stimulations in the bottom-encoding part of RL. **e** Same as panel c but for mice undergoing a rule-reversing switch ($N = 6$ mice, two-sided paired $t$ test comparing days, Det.: *$p = 0.019$; Discr.: ***$p = 1.1 \times 10^{-4}$). Performances are also tested against chance level (two-sided $t$ test, Det.: ***$p = 5.6 \times 10^{-5}$ and *$p = 0.019$; Discr.: ***$p = 7.9 \times 10^{-5}$ and Blank $p = 0.43$). **f** Same as panel d but for mice undergoing habituation to respond to optogenetic stimulations in the top-encoding part of RL. **g** Left: Comparison of the number of sessions required to reach the detection criterion between mice undergoing a rule-preserving switch and those undergoing a rule-reversing switch ($N = 7$ mice for rule-preserving and $N = 6$ mice for rule-reversing, unpaired two-sided $t$ test, *$p = 0.032$). Right: Comparison of detection performance at the end of the optogenetic habi-tuation phase (unpaired two-sided $t$ test, N.S. $p = 0.88$). Error bars: S.E.M. **h** Same as panel b but for optogenetic stimulations of area AL. **i** Same as panel d but for optogenetic stimulations of AL ($N = 7$ mice). The detection criterion was never reached during this habituation phase.

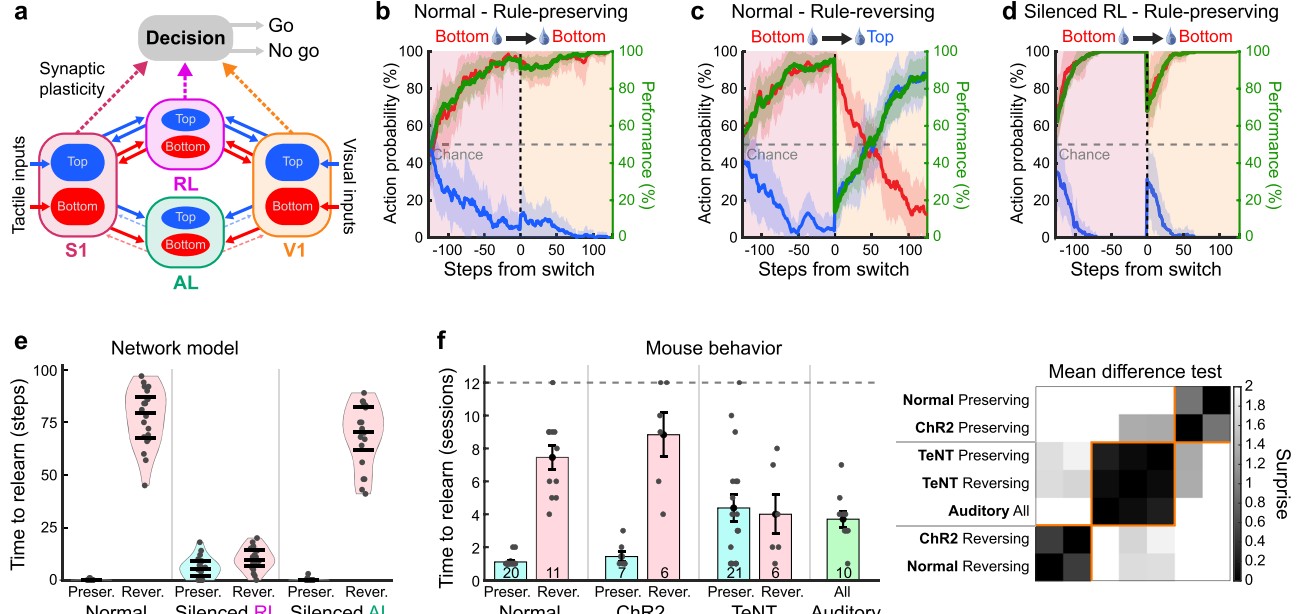

**Fig. 8 | Neural network architecture for cross-modal generalization.**
**a** Schematic of the neural network model for cross-modal generalization. Synapses projecting to the decision-computing area are the ones undergoing synaptic plasticity during sensorimotor learning. Feedback projections from area AL (dashed lines) are one tenth of the strength of feedback projections from RL (see "Methods"). **b** Action probability conditional on inputs to S1 before, and inputs to V1 after, a rule-preserving modality switch (vertical dashed line). Red: rewarded bottom stimulations, blue: non-rewarded top stimulations. Green: Discrimination performance. Shaded areas: standard deviation with $n = 20$ simulations. **c** Same as panel b after a rule-reversing modality switch. **d** Same as panel b with silenced RL. **e** Number of steps necessary to cross a performance threshold of 75% in the full network model (left) or in the model with silenced RL (middle) or silenced AL (right) after rule-preserving or rule-reversing modality switches ($n = 20$ simulations). Violin plots show the data distribution (the violin outline), while the overlaid box indicates the median (center line), interquartile range (bounds of the box), and 1.5× interquartile range (whiskers). **f** Left: Number of sessions needed to cross a performance threshold of 65% following the modality switch. Training was stopped after 12 sessions (dashed line) and mice that did not reach the criteria before this session are plotted on the dashed line. Mice numbers are indicated for each group at the bottom of the bar. Error bars: S.E.M. Right: Surprise matrix computed from pairwise unpaired two-sided $t$ test between conditions. A hierarchical clustering based on cosine similarity was used to group conditions based on surprise values.

specific cortical circuit that enables rapid generalization of sensorimotor associations learned from one modality to another, forming the neural substrate for a key component of flexible behavior.

Cross-modal generalization has been reported across numerous species including apes[3], rodents[2] and, more recently, bumblebees[4]. Early research into the mammalian brain circuits involved in cross-modal generalization underscores the necessity of different brain structures to process different types of information such as spatial, temporal, and object-related features[23,36,37]. The posterior parietal cortex (PPC) plays an important role in processing spatial information[38,39]. Although the precise anatomical and functional delineation of rodent PPC subdivisions varies across studies due to differences in nomenclature and mapping conventions, RL is generally considered one of its most posterior-lateral subregions[40,41]. Our results demonstrate that area RL is crucial for cross-modal generalization based on abstract representations of peri-personal space. Area RL displays visuo-tactile neuronal responses[19] and is implicated in whisking movements control[42]. It is strategically positioned between the primary visual cortex and the primary whisker somatosensory cortex, facilitating bidirectional information transfer between these sensory systems. Area RL possesses a retinotopic map biased towards the lower part of the visual field[24] where most whiskers are visible, and its neurons are tuned to high binocular disparity, aligning with objects in close proximity, potentially within whiskers' reach[43]. It was also implicated in optic flow processing[44] while its contribution to high-order motion computations remains debated[45,46]. Our findings unveil a fundamental functional specialization of area RL, confirming its predominant contribution in facilitating visuo-tactile coordination and sensory abstraction in the peri-personal space. Future research could

elucidate how this area may contribute to sensorimotor integration during active multimodal exploration of nearby objects, similar to multimodal limb posture encoding observed in the primate PPC[47].

Other forms of cross-modal generalization, such as object recognition, may engage different cortical regions. Indeed, areas of the dorsal stream are classically associated with spatial, attentional, and motion processing or motor guidance, while areas of the ventral stream are involved in object recognition. Previous studies have reported visuo-tactile multimodal responses in both the ventral and dorsal pathway in primates[48–50], including areas initially thought to be exclusive to visual processing like V4[51] and MT/V5[52]. Our study uncovered visuo-tactile areas belonging to the ventral or dorsal stream in mice[35,53]. Specifically, RL and A are typically associated to the dorsal stream while LI, with its significant projection to the postrhinal (POR) cortex, is associated with the ventral stream[35]. Lesions in the perirhinal cortex in rats, downstream to POR, obstruct cross-modal object recognition during spontaneous exploration[36]. Moreover, areas LI and AL have been functionally characterized as specialized for shape processing and object recognition[54,55], suggesting that they could represent a rodent homolog of the visuo-haptic subregion of the lateral occipital complex (LOtv) reported in human fMRI studies[48]. These reports suggest that the ventral pathway might contribute specifically to cross-modal object recognition. Future research focusing on objects discrimination could provide further insight into the specific role of visuo-tactile areas in the ventral stream.

Cross-modal generalization involving other sensory modalities has been observed using amodal properties such as stimulus duration or intensity[2]. Notably, rodents can generalize behaviors based on shared features between audition and vision[56,57], although the precise

circuit identity and organization required for this process remain unclear. Importantly, lesions in the posterior parietal cortex, which includes area RL, have been shown to impair behaviors relying on spatial information but not those involving non-spatial information[23]. While several other brain regions such as the prefrontal cortex or temporal associate areas may contribute to non-spatial cross-modal generalization, the exact circuit organization of these regions remains to be explored[2].

Our wide-field imaging and anatomical mapping experiments revealed a more extensive and organized network of visuo-tactile associative cortical areas than previously identified[19,35,58], where many associative areas exhibited a shared representation of vertical space between visual and whisker tactile inputs. These spatial maps extended even to primary sensory cortices through associative areas, with strong responses observed in S1 following visual stimulations despite no direct projections between V1 and S1[28]. Our data suggest that the asymmetrical propagation of activity between the visual and whisker somatosensory cortices could originate from denser populations of neurons in V1 projecting to the associative area RL compared to those in S1. However, our model suggests that this asymmetry in feedforward projections does not prevent bidirectional cross-modal generalization (Supplementary Fig. 14f). Although recent findings advise caution in interpreting cross-modal signals between primary cortical areas due to potential confusion with signals evoked by uninstructed movements[27], our results suggest that certain fundamental features, such as spatial location, indeed have shared representations across various cortical areas. These representations possibly support object or event-oriented abstract encoding[59] for cross-modal generalized learning. Organized feedback projections from associative areas to primary sensory cortices appear to play an unsuspected role for learning, in addition to their known role in shaping functional properties with contextual information[60]. This raises the question of whether the same or different feedback projections subserve these functions.

Consistent with the literature on multisensory integration[61], we observed multisensory modulations that enhanced responses to spatially congruent visuo-tactile stimuli and suppressed responses to incongruent stimuli. This response pattern could potentially result from surround suppression for incongruent inputs, mediated by local parvalbumin-positive interneurons as observed for conflicting visuo-auditory stimuli[12], while a distinct mechanism could specifically enhance responses to spatially congruent inputs. By comparing congruent to incongruent combinations, we uncovered multisensory modulation rules that shape a supramodal representation of peri-personal space. Supramodal encoding of gratings orientation was reported in neurons of the rat PPC (within its medial subdivision) in response to visuo-tactile stimuli[6] despite a lack of organized functional maps for orientation selectivity in rodents. The development of congruent supramodal tuning in single neurons and overlapping spatial maps, as observed in naive animals, likely arises from an interplay of genetically encoded connectivity biases and activity-dependent mechanisms. Biased connectivity, combined with correlated spontaneous activity in sensory areas like V1 and S1, appears to refine map alignment in regions such as mouse RL, potentially through Hebbian plasticity[62]. Similarly, in the superior colliculus, early cross-modal experiences refine coarse, overlapping modality-specific maps through Hebbian plasticity acting on sensory-evoked activity, with critical input from multisensory cortical regions[18,63]. These findings suggest that both spontaneous activity and sensory experience contribute to the maturation of multisensory integration. For example, after cataract removal, individuals initially struggle to integrate visual and haptic cues but rapidly develop optimal multisensory strategies with experience, demonstrating remarkable plasticity of these circuits even in the adult brain[64]. This adaptability raises intriguing questions about how multisensory representations in mouse RL might adjust to altered visuo-tactile contingencies, which could be experimentally tested through controlled manipulations of sensory co-occurrence statistics[65]. New experiments could provide deeper insights into the role of experience in shaping multisensory processing across developmental and adult stages.

Information generalization across modalities is crucial not only for biological brains but also for artificial systems. With the advent of multimodal large language models such as GPT-4o and Gemini, it is widely recognized that multimodality is fundamental for general artificial intelligence[66,67] and can lead to the emergence of abstract conceptual representations in machine learning systems[68]. We believe that the supramodal encoding of peri-personal space by multisensory neurons we reported could represent an early instance of the same computational mechanism leading to the emergence of conceptual representations[9,30] that form the building blocks of flexible mental representations known as cognitive maps[1]. In this light, our work represents a major step toward understanding the circuit architecture of one of the most fundamental cognitive computations: sensory abstraction.

## Methods

### Animals

All experiments were carried out in accordance with the Institutional Animal Care and Use Committee of the University of Geneva and with permission of the Geneva cantonal authorities (GE258B). C57BL/6 J wild-type mice and transgenic mouse lines were housed 2–6 mice per cage under a 12/12-h non-inverted light/dark cycle with ad libitum access to food and water. The ambient temperature in the animal facility was 23 °C and the relative humidity was maintained around 50%. Transgenic mice used for calcium imaging were obtained as a crossing between Ai148-D mice (Jackson Laboratories, stock number 030328) and Rasgrf2-2A-dCre mice (Jackson Laboratories, stock number 022864). Ai148-D mouse line is a Cre-dependent reporter line containing a gene encoding the calcium indicator GCaMP6f at the Igs7 locus. Exposure to Cre recombinase through viral vector injections or crossing with Cre-expressing mice resulted in expression of GCaMP6f. Rasgrf2-2A-dCre mouse line expressed a trimethoprim-inducible Cre recombinase directed by endogenous Rasgrf2 promoter/enhancer elements. When induced, Cre recombinase activity is observed in cortical layers 2/3 and other scattered cells of the cortex, hypothalamus, thalamus, and midbrain. Trimethoprim (TMP) i.p. injections were performed for 5 consecutive days at least two weeks before any surgical intervention (0.25 mg/g of body weight diluted in DMSO and 0.9% NaCl). As a result, TMP injected crossed Ai148D x Rasgrf2-2A-dCre mice expressed GCaMP6f in cortical layer 2/3. For optogenetic experiments, we used Ai32 transgenic mice that express Channelrhodopsin-2 (ChR2) fused to enhanced Yellow Fluorescent Protein (eYFP) in a Cre-dependent manner, enabling fast neuronal activation in vivo through blue light illumination (Jackson Laboratories, stock number 012569). Males and females aged 2–5 months and weighing approximately 20–25 g were used for all experiments with no clear sex differences observed.

### Viral vectors and markers

For anterograde labeling, AAV2.CAG.GFP (Addgene, #37825-AAV2, titer: $7 \times 10^{12}$ vg/mL) and AAV2.CAG.tdTomato (Addgene, #59462-AAV2, titer: $4 \times 10^{12}$ vg/mL) were used. The volume of viral vectors injected was around 50–75 nL in each site at a depth of approximately 400–500 μm from the surface of the cortex. Brains were collected 3 weeks after the injections. For retrograde labeling, Cholera Toxin subunit B conjugated to Alexa Fluor 555 (Invitrogen, reference number C22843) or Alexa Fluor 647 (Invitrogen, reference number C34778) diluted in PBS were injected (-50 nL in each site, 400–500 μm below the surface). Brains were collected 10 days after the injections. For silencing experiments, AAV-DJ.CMV.eGFP-2A-TeNT viral vectors were injected in the region of interest (Wu Tsai Neurosciences Institute,

reference number GVVC-AAV-70) at 3 different depths along the cortical column (800 μm, 500 μm and 200 μm respectively). Behavioral protocols started at least one month after the injections were done. For optogenetic experiments, Ai32 mice expressing Cre-dependent ChR2-eYFP were injected with AAV1.CamKII0.4.Cre.SV40 (Addgene, #105558-AAV1, titer: $3.5 \times 10^{12}$ vg/mL) in three different locations along RL and two different depths (250 μm and 500 μm) to express ChR2-eYFP in most of this area.

## Stereotaxic surgeries

Pain management was first performed by administering the opioid Buprenorphine subcutaneously (0.1 mg/kg) before starting the surgery. Mice were anesthetized inside an induction chamber with 3% isoflurane mixed with oxygen and then fixed on the stereotaxic apparatus (Model 940, Kopf). A custom-made nose-clamp has been adapted to the apparatus to maintain the position of the animal, allowing head rotation. Body temperature was constantly monitored through a thermic probe and adjusted to ~37 °C via a heating pad placed below the mouse (DC Temperature Controller, FHC). Breathing rate was regularly monitored by visual inspection. Ophthalmic gel (Vitamin A, Bausch Lomb) was applied on both eyes to ensure protection from light and prevent them from drying out. A local anesthetic was injected under the skin of the head before the surgical incision (mix Lidocaine/Bupivacaine, 6 mg/kg and 2.5 mg/kg respectively). During the surgery, mice were anesthetized with a constant isoflurane level lowered around 2%. At the end of the surgery, an anti-inflammatory was also administered subcutaneously (Carprofen, 7.5 mg/kg) and animals were warmed with a heating lamp for at least 15 min until recovery from anesthesia. A second anti-inflammatory (Ibuprofen, Algifor) was added to drinking water for 3-days post-op and the weight was checked daily to ensure that weight was kept above 15% of the original weight prior to the surgery. All animals were implanted with a cranial implant. After removing the skin and tissues on top of the head, the skull was cleaned, dried and thinned. The mouse head was tilted approximately at a 30° angle, ensuring better access to the left hemisphere. A custom-made metallic implant was placed on the top of the skull using a custom-made holder. It was then fixed with a layer of glue (Loctite 401, Henkel) and additional layers of dental acrylic (Pala, Kulzer) to ensure the implant is securely attached. Dental acrylic was covered with black nail polish to prevent light contamination from visual stimuli during imaging experiments. After at least 3 days of recovery, mice could undergo additional procedures. Animals that underwent imaging sessions were implanted with a glass window composed of a top round cover slip of 7 mm diameter and two superimposed 5 mm diameter cover slips (CS-7R and CS-5R, Multi Channel Systems). The three concentric cover slips were glued together with UV glue (Optical Adhesive n°68, Norland). The craniotomy was the size of the smaller window and was drilled above the posterior part of the dorsal cortex. Before removing skullcap, we used a custom-made perfusion chamber with saline solution (NaCl 0.9%) to rinse the craniotomy continuously and reduce bleeding stains. A custom-made holder with air-suction was used to hold and position the window. The cranial window was gently brought down until it was in contact with the brain. The window was then fixed with UV glue, super glue and dental acrylic. For stereotaxic injections, glass pipettes (5-000-2005, Drummond) were pulled (Model P-97, Sutter Instrument Co) and broken to obtain a tip of ~10–15 μm inner diameter. Pipettes were further beveled to create a sharp tip to avoid cortical damage during insertion. Injection sites were determined using functional mapping. Injections were done using a single-axis oil hydraulic micromanipulator (R.MO-10, Narishige). The pipette was slowly inserted inside the cortex until reaching the desired depth. Viral vectors or other reagents were injected at a speed of ~2nL/s. When the whole volume was injected, we waited 5 min with the pipette in the same position before gently removing it.

## Behavioral training

A Matlab custom-made graphical user interface (GUI) was developed from the Matlab App Designer to control the behavioral tasks and monitor performance. The GUI allowed real-time visualization of the animal's performance and online modification of the parameters (e.g. stimuli parameters, punishment duration, stimuli proportion). Animals underwent water restriction 2–4 days before the training started and were handled every day by the experimenter for at least 10 min. During the pre-training phase, mice were habituated to head-fixation and placed on the setup. In the first session, only Go stimuli (i.e., stimuli for which licking responses are rewarded) were presented and rewards were delivered automatically regardless of mouse actions. Go trials proportion was progressively reduced as No go trials (i.e., trials featuring the stimulus for which licking responses were punished) were added in the following sessions and mice were required to lick to obtain rewards during Go trials. Mice were trained once a day, every day at the same hour. Eight different behavioral tasks (two tactile tasks, two visual tasks, two auditory tasks and two optogenetic tasks) were used for our experiments (Figs. 1, 2, Supplementary Figs. 1, 7), all following a Go/No go discrimination paradigm. In the tactile tasks, two whiskers ("top" whisker B2 and "bottom" whisker C2) were inserted inside glass capillaries each attached to a piezo actuator that could create a small deflection of about 1 mm along the rostro-caudal axis. These deflections were sinusoidal pulses and lasted 0.2 s. In one version of the task, B2 whisker stimulations were associated to a reward (Go trials). In the other version, C2 whisker stimulations were the Go trials. In the visual tasks, two drifting squares (a "top" square and a "bottom" square, relative to the midline in the mouse visual field) were presented on the screen. In one version of the task, top square stimulations were the Go trials while in the other version of the task, bottom squares stimulations were the Go trials. The auditory task followed the same structure as the other tasks but using two short pure tones of 6 kHz and 12 kHz as Go and No go stimuli, respectively. These tones were delivered from a speaker located next to the mouse on the same side as the visual and tactile stimuli. The optogenetic tasks were performed by directly stimulating subregions of RL or AL corresponding to top or bottom multimodal representation found with calcium imaging, using shaped blue light patterns. In one version of the task, stimulating the top region was associated with a reward, while in the other version, it was the bottom region. The total trial duration of a single trial was 4 s: after a 2 s quiet window (during which licking resulted in trial abortion) the stimulus was presented, and the mouse was allowed to lick during a 2 s response window. During Go trials, the mouse could obtain a water reward upon licking the spout following the stimulus (Hit trials). Failure to lick would result in a Miss trial. During No go trials, the mouse had to refrain from licking (Correct rejection trials, CR) or it was punished with a time-out of 10 s (False Alarm trials, FA). Some trials were presented without any stimulus (catch trials). If the mouse licked during catch trials, no time-out was applied. If mice were too compulsive (i.e. licking during the quiet window was too frequent), a 10 s time-out early lick punishment could also be applied. The proportion of each trial type was the following: Go trials = 30%, No go trials = 50%, Catch trials = 20%. All tactile stimuli were generated through Matlab data acquisition toolbox controlling a piezo actuator (Bimorph bendor piezo actuator PB4NB2S, Thorlabs) through a National Instrument card. All visual stimuli were generated using Matlab and PsychToolBox. Stimuli were presented on a gray background through a LCD monitor (20 × 15 cm, pixels, 60 Hz refresh rate, Pi-shop) positioned 10 cm from the eye, with a 30° angle to the right of the midline. The screen was also tilted with a 30° angle along the horizontal plane to match the mouse head angle with the intent of roughly aligning the bottom and top parts of the screen to the resting position of the C2 and B2 whiskers in the mouse visual field. Stimulations during behavior consisted of black

squares moving through the screen in the rostro-caudal direction (bar width was 12.5°, stimulus duration was 175 ms, speed was 500°/sec), on a gray background.

## Wide-field microscopy

We used a custom-made wide-field epifluorescence microscope setup[46] including a sCMOS camera (ORCA-Flash4.0 V3, Hamamatsu). Magnification was achieved through a 0.63X C-mount camera adapter for Olympus Microscopes. The field-of-view size was 5.6 mm × 5.6 mm. The camera and adapters were mounted on a base allowing vertical movement with manual focus. LED white illumination (740 mW, 1225 mA, Thorlabs) could be controlled via a T-Cube LED driver (LEDD1B, Thorlabs). Filter cubes could be changed for different types of imaging. For imaging GCaMP6f, GFP excitation, emission and dichroic filters were used. For intrinsic optical signals imaging, Cy3/5 excitation, emission and dichroic filters were used. An objective (MVX Plan Apochromat with 2x, Olympus) was attached to the microscope base. The somatotopic mapping protocol consisted of repetitive rostro-caudal pulsatile deflections (-1 mm amplitude) of either B2 or C2 whiskers for 50-80 trials each followed by a quiet window. The retinotopic mapping protocol consisted of drifting bars. A contrast reversing checkerboard was presented within the bar to better drive neural activity (0.04 cycles/° of spatial frequency and 2 Hz of temporal frequency). In each trial the bar was swept in the four cardinal directions: left to right, right to left, bottom to top, and top to bottom. Single trials were repeated 20 to 40 times. For anatomical experiments, only C57BL/6 J mice were used. To image the intrinsic optical signal, we used longer tactile stimulations and slower visual stimulations. Mice were fixed on the platform and anesthetized during the procedure with isoflurane level lowered than 1%. Body temperature was monitored with a probe and adjusted to 37 °C using a heating pad (DC Temperature Controller, FHC). For all the other experiments, we used Ai148-D x Rasgrf2-2A-dCre mice and calcium imaging. The visuo-tactile sparse noise protocol consisted in combinations of visuo-tactile stimuli: three whisker conditions (C2 whisker stimulation, B2 whisker stimulation and no whisker stimulation) and nine vertical positions of a moving square similar to the one used for the visual task. The ninth position corresponded to a no visual stimulus condition (i.e. blank screen). Together, all visual stimuli spanned approximately $8 \times 12.5° = 100°$ in the vertical space. Visual and tactile stimuli were either presented alone (unisensory conditions) or together (multisensory conditions). When presented together, onsets were either synchronous or delayed (0.15 s delay, visual leading tactile stimulus). In total, $3 \times 9 \times 2 = 54$ different combinations were presented in pseudo-random order with a 1 s inter-stimulus interval. The full sequence was repeated 60 times. Total sparse noise protocol duration was approximately 1 h. At the beginning of each recording, we took a picture of the cranial window with blood vessels pattern on focus as reference image. The focus was then set at -300 μm below the surface to maximize signal collection. Light was adjusted to prevent saturation. Before each imaging session, the window was cleaned with 70% ethanol and eyes hydrated with mineral oil.

## Two-photon microscopy

The two-photon microscope was custom-made (INSS Company). It consisted of a femtosecond laser with wavelength range 690–1040 nm (Tunable Ti:Sapphire with dispersion compensation MaiTai DeepSee, Spectra Physics) whose beam was directed with Resonant/Galvo scan mirrors and the emitted signals were detected by 2 GaAsP amplified PMTs (PMT2101/M, Thorlabs). Imaging was performed through a 16x Nikon 0.80 NA objective and using ScanImage (Vidrio Technologies). Images were acquired at approximately 30 frames per second. Two-photon calcium imaging during the visuo-tactile sparse noise protocol described in the wide-field section was performed on Ai148-D x

Rasgrf2-2A-dCre mice. After cleaning the cranial window with 70% ethanol, a hydrophobic chamber was made between the head plate and the imaging platform using liquid plastic (Smooth-Cast 325, Smooth-On) and the objective was immersed in distilled water. We ensured no polluting light could reach the objective by covering it with a dark protective sleeve and by turning off the light in the room. At the beginning of each recording, an anatomical picture of the field-of-view was taken using a CCD camera. When switching the microscope in two-photon mode, we took an image of the surface blood vessels at magnification ×1 and ×1.5 for further realignment. For each location, we typically imaged at three different depths between 100 μm and 300 μm below the surface. Each mouse underwent no more than two imaging protocols per day.

## Histology

Mice were anesthetized with 3% isoflurane and euthanized with i.p. injection of Pentobarbital (Eskornarkon, 150 mg/kg) and Buprenorphine (0.1 mg/kg). They were then transcardially perfused using a peristaltic pump (ISM829, Cole-Parmer) with 0.01 M PBS, pH 7.4 for 2 min, and then 4% paraformaldehyde (PFA) in phosphate buffer (PB; pH 7.4) for 3 min. The brains were post-fixed at 4 °C in PFA for 48 h and then transferred into PBS. For anatomical experiments of Fig. 4, the brains were post-fixed at 4 °C in PFA for 2 h and washed 3 times for 15 min in PBS. The left cerebral hemisphere was separated from the right hemisphere and subcortical parts were removed with a spatula. The left hemisphere was flattened between glass slides and kept flattened at 4 °C in PFA for 12 h. After PBS washing, flattened hemispheres were embedded in agar gel 4% before cutting slices of 70 μm thickness with a vibratome (VT 1000 S, Leica).

## Immunohistochemistry

Flat brain sections were incubated with slight agitation (40 rotations/min) for 2 h at room temperature in a saturation/permeabilization solution containing a mix of 5% BSA and NGS, 0.3% triton X-100 and PBS. Brain sections were then incubated with slight agitation (40 rotations/min) overnight at 4 °C with a rat anti-muscarinic acetylcholine receptor M2 (M2 AChR) primary antibody (1:500, Sigma-Aldrich, reference number MAB367) in the same blocking solution. Sections were washed 3 times for 15 min in PBS before a 2 h incubation with either a donkey anti-rat secondary antibody conjugated with Alexa Fluor 488 (dilution 1:500, Invitrogen, #A21208) or with a goat anti-rat secondary antibody conjugated with Cy5 (dilution 1:500, Invitrogen, #A10525) in blocking solution at room temperature without agitation. Hoechst solution (dilution 1:1000, Invitrogen, # 33342) was used for fluorescent nuclear counterstaining. Slices were incubated for 10 min in that solution with slight agitation then washed 2 times for 5 min in PBS.

## Histological imaging

Slices were mounted on Superfrost microscope slides (Epredia) with mounting medium (Fluoromount) and covered with 24 × 50 mm coverslips (Menzel-Gläser). All photomicrographs were taken using a Zeiss Axio Scan.Z1 or a Leica Stellaris 5 microscope at the bioimaging platform of the University of Geneva.

## Wide-field calcium imaging analysis: retinotopic and whisker mapping protocols

Responses to drifting checkerboard stimuli were averaged across trials for each condition and converted in df/f using a pre-stimulus time window as baseline. Azimuth and elevation preference maps were then computed[24]. To segment visual areas, azimuth and elevation maps were combined to generate a visual field sign map. The sign map was computed as the sine of the difference between the vertical and horizontal retinotopic gradients for each pixel. Somatotopic maps were

obtained by averaging the response df/f across B2 or C2 stimulation trials over a short time window of 200 ms.

## Wide-field calcium imaging analysis: sparse noise protocol

After an initial down sampling to a resolution of 100 × 100 pixels by bicubic interpolation, 50 Hz framerate videos acquired during the sparse noise protocol underwent pixelwise notch filtering ($f_0 = 12$ Hz, $f_w = 6$ Hz) and discrete wavelet transform (DWT) detrending (setting lowest frequency approximation coefficients to zero in a 6-level decomposition) to remove artifacts and low-frequency drifts. After these initial preprocessing steps, videos were z-scored using as reference distribution the ensemble of all pixel values corresponding to blank stimuli (i.e. trials with no visual and no tactile stimulation). Next, trials with a high baseline activity (>=0.5 z-score on average, corresponding to a strong positive fluctuation of spontaneous activity preceding the stimulus) were discarded and z-score and baseline subtraction were computed again including only remaining trials. This was meant to better separate local responses to sensory stimulations from ongoing spontaneous activity. Starting from the z-scored data obtained in this way, response surprises (i.e. $-\log10(p_{t\text{-test}})$) across trials were computed for each time point and each stimulus to define spatial responsivity masks. Average visual or tactile top or bottom response movies were obtained by averaging the median responses across trials to the relevant stimuli. For visual "top" all stimulation conditions (either unimodal or multimodal) in which the visual stimulus was present in grid positions 7 or 8 were used whereas for visual "bottom" all visual conditions including positions 3 or 4 were used. Retinotopic and somatotopic maps of vertical space (Fig. 3b,c and Supplementary Fig. 6g) were obtained by taking the difference between average visual or tactile top and bottom response maps. Modality preference maps (Fig. 3d and Supplementary Fig. 6g) were obtained by computing pixel-by-pixel the difference between the maximum response for visual and tactile stimuli. Multisensory enhancement maps (Fig. 3f and Supplementary Fig. 6g) were obtained by computing pixel-by-pixel the difference between the maximum unisensory response between visual and tactile stimuli (predicted response) and the multisensory response (measured responses). For all the difference maps mentioned above the input maps were independently normalized to the [0,1] interval before taking the difference (ensuring their range to be bound in the [-1,1] interval). Retinotopy-somatotopy spatial coherence maps (Fig. 3e and Supplementary Fig. 6g) were obtained by convolutionally (stride = 1 patch size = 8 pixels) computing the correlation-based similarity of matching patches of the two different maps. The result was then scaled by the average peak-to-peak range in that patch (to diminish contribution from pixel with very low response magnitude). Comparison of grand average maps at single pixel level (Fig. 3g,h) was performed on responsive pixels only (exceeding a surprise threshold corresponding to $p_{t\text{-test}} <= 0.1$ for either visual or tactile responses). We compared pixels with low and high modality preference (i.e. below or above 85% quantile of the absolute value distribution within the responsive region) or with low and high coherence (i.e. below or above 15% quantile of the absolute value distribution within the responsive region). Grand average maps displayed in Fig. 3b–f are obtained by averaging frames of the corresponding grand-average movie (obtained by realigning single mouse movies to the common reference atlas and then averaging) over different time windows: frames 18 to 20 for the somatotopy (i.e. 0.16–0.20 s after stimulus onset); frames 25 to 27 for the retinotopy (i.e. 0.30–0.34 s after stimulus onset); and frames 28–30 for multisensory modulation (i.e. 0.36–0.4 s after stimulus onset). These windows were chosen to be roughly centered on the peak of these signals. Grand average modality preference and spatial coherence maps displayed in Fig. 3d,e were computed on maps obtained above (average z-score maps or position selectivity maps, respectively). Signal time courses displayed in Supplementary Fig. 7 were obtained by integrating grand-average movies across time-varying regions of interest (ROIs). These ROIs were specifically defined to track the activity 'bump' evoked by stimulation in each area. To prevent bias in estimating the response—particularly in areas with higher cortical magnification factors, which could lower the average by including many non-responsive pixels—these masks were created by intersecting a static mask, which selects each area based on the fitted reference atlas, with a dynamic responsivity mask. This dynamic mask selects, for each movie frame, only those pixels that across mice show an average maximum response surprise, exceeding a threshold (corresponding to a $p$-value $\leq 0.01$ across conditions). To measure angular mismatch between retinotopic and somatotopic maps we computed vertical retinotopy gradient vectors from the grand-average version of these maps (using Matlab "gradient" function with a scale of 4 pixels followed by a Gaussian smoothing of gradient components). Then a region of interest overlapping with RL was defined and we computed the average angular difference between the two gradient fields.

## Atlas fitting and registration

The reference Allen Mouse Brain atlas[69] was first tilted sideways at 30° and then projected to match the skull tilt in our experiments and therefore the layout of areas in the dorsal cortex. The resulting atlas was manually fitted to the wide-field imaging field-of-view for each mouse by visually aligning the atlas boundary lines to reproducible landmarks from functional maps. Both the maps obtained from the sparse noise protocol (i.e. vertical retinotopy and somatotopy difference maps) and the maps obtained from the whisker and retinotopic mapping protocols were used to register the atlas. Landmarks used include the outlines of the sign map regions, the position of C2 and B2 whisker activity bumps in S1 and S2, the reversal of vertical retinotopy at the boundary of each visual area (see Supplementary Fig. 6 for example). To relate microscale and macroscale functional properties measured respectively by two-photon and wide-field imaging experiments, we also manually reconstructed the position of each two-photon field-of-view (FOV) in the frame of reference with the atlas fitted to the wide-field maps. To do so, we aligned the blood vessels pattern visible in each two-photon FOV with the one visible in the wide-field cortical image (as shown in Supplementary Fig. 9f). To compare data across mice and compute grand averages, we developed a Matlab pipeline to robustly realign atlases to one another. This pipeline was based on iterative application of the image registration algorithm implemented by Matlab function "imregtform" (considering "rigid" transformations in "monomodal" mode) to stack images depicting atlas boundaries. This pipeline enabled us to obtain the rotation and shift required to register all maps and neuron positions to the frame of reference of a common atlas. For the anatomical tracing experiments (Fig. 4b–d), a similar processing was performed with ImageJ through registration of atlases fitted using the M2 AChR staining. Brain from all mice with similar injections could then be averaged together. To compare the somatotopy and retinotopy of axonal projections or projecting neurons with the functional topographic maps obtained in wide-field we brought them in a common reference frame by registering a crop of the M2 AChR-fitted atlas to the common wide-field/two-photon atlas (by finding the registering affine transformation with Matlab functions "cpselect" and "fitgeotrans").

## Two-photon calcium imaging analysis: pre-processing and functional cell-types

To extract time-varying somatic GCaMP6f calcium signals, we used the Suite2p toolbox[70]. Neuropil contamination was corrected by subtracting the fluorescent signal from a surrounding ring $F_{Surround}(t)$ from somatic fluorescence: $F(t) = F_{Soma}(t) - \alpha * F_{Surround}(t)$ with $\alpha = 0.7$. Neuropil-corrected fluorescence signals $F(t)$ were then converted in z-score by subtracting from each trace the mean value and dividing by the standard deviation of $F(t)$ over the last 0.2 s of the baseline window preceding the stimulus onset (pooling across all trials). Using these

z-scored fluorescence traces we computed response surprises (i.e. -log10($p_{z\text{-test}}$)) across trials for each stimulus and for each time sample. Similarly, we computed the coefficient of variation (CV) across trials quantifying response variability. A neuron was considered reliably responsive if its response exceeded a stringent surprise threshold $Surprise_{th} = 8$ (i.e. $p_{z\text{-test}} <= 10^{-8}$) while remaining below a coefficient of variation threshold value $CV_{th} = 4$ for at least 4 consecutive time bins in any stimulus condition. This responsivity criterion was requested for each neuron to be included in subsequent analyses. For each neuron included, we used a more inclusive criteria to characterize the extent of their tuning properties. Indeed, neurons exceeding a less stringent threshold $Surprise_{th} = 2$ (i.e. $p_{t\text{-test}} <= 0.01$) while remaining below the same coefficient of variation threshold at any time bin of a visual, tactile or visuo-tactile condition were considered "responsive" to these conditions. The responsivity pattern of each neuron was used to classify it in one of four functional classes: (1) neurons responsive in visual but not tactile conditions were termed "visual neuron"; (2) neurons responsive in tactile but not visual conditions were termed "tactile neuron"; (3) neurons responsive in both tactile and visual conditions were termed "visuo-tactile neuron"; (4) neurons responsive only in visuo-tactile conditions (both visual and tactile stimuli presented together) were termed "gated neuron". Only visuo-tactile neurons displaying significant responses in both modalities were used for the analyses displayed in Fig. 5g–j. Measured multisensory responses were compared to a "max model" of multisensory interaction[71] (as in Fig. 5b and Supplementary Fig. 11b). Predicted visuo-tactile z-scored traces for each multisensory stimulus condition were computed trial-wise by taking the max (for each time bin) between the response to the same visual and tactile stimulus presented alone (i.e. the corresponding unimodal conditions) and subsequently averaging across trials. Average responses (predicted or measured) were computed over a response window spanning from 0.2 s to ~0.733 s following stimulus onset for each stimulus and each responsive neuron.

**Two-photon calcium imaging analysis: Computation of indices**

We computed the "multisensory modulation index" (MI) as the normalized difference between predicted and measured visuo-tactile responses at every time bin for every stimulus condition. To summarize the intensity of multisensory modulation of each neuron across stimulus conditions over time we computed the min-max range of the integral of MI values over the same time window mentioned above (i.e. from 0.2 s to 0.733 s) across all valid conditions. Valid stimulus conditions comprised significant responses as well as all multisensory conditions containing at least one unisensory stimulus to which the neuron is responsive and displaying significant multisensory modulations (i.e. with $p < 0.05$ of a bootstrap-$t$ test for MI value for at least one time bin, see "quantification and statistical analysis" section below). Preferred vertical position in the tactile or visual space of each neuron was quantified as the center-of-mass of these average responses along the corresponding dimension of the stimulus grid on visuo-tactile (measured or predicted) conditions. This center-of-mass was converted into a "position preference index" ranging from +1 for neurons tuned to the top visual/B2 whisker stimulus to -1 for neurons tuned to the bottom visual/C2 whisker stimulus. This index characterizes the tendency of each neuron to respond more to stimuli located at one particular end (i.e. "top" or "bottom") of the spectrum of stimulus positions presented in each modality (as in Fig. 5c–g and Supplementary Fig. 10a, b). The sharpness of positional tuning of neurons was computed as the inverse of the best sigma parameter of a Gaussian fit of the average response to each visual position or as the absolute value of the tactile position selectivity index described above. This selectivity index was computed on the visuo-tactile condition of max response (both in the measured or predicted case). These indices ranged from 0 for neurons responding equally to all positions (i.e. completely positionally untuned) to 1 for neurons responding to only

one position (i.e. maximally positionally tuned). The values of positional preference and selectivity indices shown in Fig. 5g–j are averaged over both delay conditions (i.e. with or without delay between tactile and visual stimuli, see Supplementary Fig. 7). Neurons were labeled as ventral or dorsal by running a k-means clustering algorithm (with $n_{centroids} = 2$) on the spatial distribution of visuo-tactile neurons over the surface of the dorsal cortex (Fig. 5e).

**Two-photon calcium imaging analysis: correspondence with wide-field**

To quantify the correspondence between single-neuron properties distributed across the cortex and the wide-field maps, we created smoothed spatial maps from 2D histograms that depict the dominant response properties at each cortical location. We generated retinotopic and somatotopic maps from two-photon data (Fig. 5c, d and Supplementary Fig. 10a,b) by smoothing a 2D normalized histogram of neurons with positional preferences (top-preference index ≥0.05, bottom-preference index ≤0.05) using a Gaussian kernel (σ-3% of total wide-field FOV) and then calculated the difference between these maps. For modality preference maps reconstructed using two-photon data (Supplementary Fig. 10c), we applied a Gaussian smoothing (σ-3% of FOV) to 2D normalized histograms that describe the distribution of unimodal visual and tactile neurons, calculating the normalized difference between them. To map multisensory modulation (Supplementary Fig. 10d), we applied Gaussian smoothing (σ-3% of FOV) to a MI-range-weighted histogram showing the spatial distribution of multisensory modulations, then normalized this against the unweighted distribution that includes unresponsive neurons. We included unresponsive neurons to properly capture the relative density of multisensory modulation in the local population. To quantitatively assess the match between two-photon reconstructed maps and the wide-field maps, we computed a Pearson correlation coefficient restricted to the pixels with good two-photon coverage (thresholding the map shown in Supplementary Fig. 9g. at a threshold of 2000 neurons/mm2).

**Two-photon calcium imaging analysis: decoding analysis**

To test whether the neuronal representation observed across the dorsal cortex of naïve mice could support cross-modal generalization observed in our behavioral experiments, we adopted a decoding approach. We trained a Bayesian classifier to infer stimulus category (top or bottom) from single-neuron responses in the tactile modality and then tested the accuracy on the visual modality. This decoder was inspired by original works introducing neuronal population probabilistic codes[72,73] and is similar to the approach used in previous work[20]. We built feature vectors representing the response of each neuron to each stimulus category (i.e. "top" and "bottom") in each modality. The feature vectors were defined, for each neuron, as the z-score response integrated over a 0.2 s–0.733 s time window in every trial of the most responsive stimulus position pertaining to each category (visual top = positions 5 to 8, visual bottom = positions 1 to 4, tactile top = B2, tactile bottom = C2). The response feature vectors were used in a trial-wise fashion as predictors to infer the "top" or "bottom" labels corresponding to the location of the stimulus delivered to the animal. We incorporated fourfold cross-validation to mitigate overfitting and assess the classifier's accuracy outside the training dataset. Trials were partitioned into four batches, three of them for training and one for testing. The decoding was assessed both at the single neuron level ($n_{pool} = 1$ and averaged over all neurons of a given class to get the values displayed in Supplementary Fig. 11d, e) as well as on pools of neurons of increasing size ($n_{pool} = [1,50]$ to get the curves displayed in Supplementary Fig. 11c). During population decoding, we repeated the procedure 40 times, each time randomly resampling the neurons in the decoding pool without replacement to ensure that the results were not dependent on a specific selection and partitioning of neurons.

Decoding accuracy (defined as the fraction of trials with correct predictions) was averaged across resampled data and cross-validation splits. Prior to classification, each neuron's response features within the population vector were individually normalized to a 0–1 range across trials. Additionally, trials were randomly reordered, maintaining their corresponding correct labels, to eliminate noise-related correlations. In this probabilistic framework the "training" consists simply in computing the average tuning curves (i.e. stimulus-triggered average) using all trials included in the training set and taking the logarithm of this vector. The subsequent "testing" of the decoder requires to generate a prediction. Predictions were made by calculating each test trial's feature vector log-likelihood for the 'top' or 'bottom' stimulus, then assigning the label with the highest log-likelihood as the predicted outcome. The log-likelihood can be obtained by taking a matrix product between the matrix storing the log of tuning curves computed at the "training" stage and the current population feature vector (after subtracting a corrective term proportional to the sum of all tuning curves to account for any representational bias in the decoded population). Tactile decoding performances represented in Supplementary Fig. 11c, d, e corresponded to the average cross-validated test accuracies on the same modality on which the Bayesian classifier was trained (i.e. tactile).

## Behavioral analysis

To quantify mouse behavior over time within single sessions we isolated different stimulation conditions (i.e. trials with top stimuli, trials with bottom stimuli and Catch trials) to compute a smoothed conditional lick probability over a sliding window of 60 trials. We calculated the conditional probabilities of licking by dividing the number of trials with a mouse lick response to a specific stimulus by the total presentations of that stimulus within the moving time window. These curves are the ones displayed throughout the paper (Figs. 1, 2, 6, 7 and Supplementary Figs. 1, 3, 4, 12, 13) and labeled "Top", "Bottom" and "Spont". From Go and No go lick probabilities (defined by the task rule to coincide with top or bottom), we also computed a "Discrimination performance" metric as the percentage of correct trials for discrimination. Specifically, we defined behavioral performance as the average between the rate of correct choices upon delivery of the Go stimulus (i.e. Hit rate = conditional lick probability for Go trials) and the rate of correct rejects upon delivery of the No go stimulus (i.e. Correct Reject rate = 1 - False Alarm rate where False Alarm rate = conditional lick probability for No go trials). This way of computing performance makes it insensitive to the proportion of Go and No go trials. Following a similar logic, we defined a "Detection performance" as the maximum between the Go detection rate (i.e. average of Hit rate and Catch Correct Reject rate) and the No go detection rate (i.e. average of False Alarm rate and Catch Correct Reject rate). Behavioral states were classified as previously described[20]. The end of each session was determined as the trial following peak performance where the conditional lick probability for Go trials dropped below 66% of its maximum value. Conditional lick probabilities and performance per session were computed only during engaged trials of each session (i.e. discarding the part of the session during which conditional lick probabilities dropped due to task disengagement). Switch-aligned performance trajectories shown throughout the paper (Figs. 1, 2, 6, 7 and Supplementary Fig. 1, 2, 4, 13) were obtained realigning the curve of average session performance and conditional lick probabilities around the session of task switch. Accompanying bar plots display the difference of discrimination and detection performance between the last session before the switch and the first session after the switch. In Supplementary Fig. 3, we plotted the average conditional lick probabilities and the performance time course across mice that underwent whisker removal control sessions. These average curves were obtained by dividing each session in three

chunks: "before", "during" and "after" whisker removal and averaging them separately after a stretching interpolation to match their duration. Accompanying bar plots display the average discrimination and detection performance in these three chunks (excluding boundary timepoints to avoid contamination due to the temporal smoothing induced by the sliding window we used to compute the traces). Learning rates following the switch (Fig. 2c and Fig. 6l) were obtained measuring the slope of the best fitted line for 3 consecutive sessions after switch for all mice in each group. We excluded mice that did not show a drop of performance to chance level after the switch as they successfully generalized from one modality to the other. The relationship between the TeNT-P2A-GFP expression overlap with each cortical area and the cross-modal generalization performance impairment in rule-preserving conditions was assessed using Pearson coefficient of correlation. Specifically, for each mouse we calculated the Pearson coefficient of correlation between two vectors: 1) the vector representing the fractional overlap of TeNT expression binary masks (displayed in Supplementary Fig. 12) and each atlas-defined cortical area, and 2) the vector representing the change in performance following modality switch (performance averaged over 3 consecutive sessions after switch). This analysis was performed for each cortical area (examples for areas RL and AL are shown in Supplementary Fig. 12f) and displayed as a correlation map in Fig. 6e. For the reverse correlation approach, we computed the difference between the average of all TeNT-injected mice GFP fluorescence binary masks and the one of mice displaying impairment of cross-modal generalization ability (defined as mice performing below 57.5% of correct trials on average in the 3 sessions following a rule-preserving modality switch). This difference took null or negative values in areas where TeNT expression did not affect the behavior and positive values in regions producing behavioral impairment (i.e. regions where TeNT expression is enriched in the impairment-triggered ensemble compared to the ensemble including all TeNT-expressing mice).

## Quantification of TeNT expression and ventral/dorsal grouping

Wide-field fluorescence images of dorsal cortex displaying the pattern of expression of TeNT-P2A-GFP in each injected mouse were first pre-processed to remove the blood vessels from the surface. This was necessary to obtain a smooth, non-occluded estimate of the spatial profile of GFP expression across the surface of the cortex. To do so we first segmented the blood vessels with adaptive thresholding (using Matlab function "adaptthresh") and then inpaint them by local interpolation (using Matlab function "regionfill") (Fig. 6g–i and Supplementary Figs. 12a–d and 13b). Next, we computed a median-subtracted map of df/f calculated with respect to a manually annotated region of interest on the border of each window (selected to correspond to a fluorescence profile stable around baseline level, usually at the farthest end of the window with respect to the injection site). To compare images across different mice and account for variations in imaging conditions, we normalized saturation by clipping the values in each map to the minimum peak value observed across all mice. After normalizing these resaturated df/f maps to 1 we computed their 0.90 contour (i.e. the expression "patch") to define the binary TeNT-P2A-GFP expression mask to be used in subsequent analyses to evaluate the spatial extent of toxin expression. The overlay of these masks for all mice is displayed in Supplementary Fig. 12e. Next, TeNT-expressing mice undergoing rule-preserving modality switch were subdivided in "dorsal" and "ventral" subgroups by computing a d-prime measure (distance normalized by patch size) with respect to the two multimodal visuo-tactile neuron clusters found by k-means in the two-photon analysis of Fig. 5e. Neurons outside the region of low d-prime with patch centroid falling respectively within AL/LI or A/RL region were categorized as ventral

and dorsal, respectively (Fig. 6g–i showed the overlay of the binary masks together with expression centers as dots over the common atlas). The same procedure was followed to validate the targeting of TeNT injections to the dorsal region in mice undergoing rule-reversing modality switch (Supplementary Fig. 13b). To quantify potential differences in the size of TeNT-expressing cortical patches between the "ventral" and "dorsal" subgroups of animals, while accounting for the surface obstructed by dental cement at the edge of the cranial window, we performed a Gaussian fit on the normalized df/f images mentioned above. A 2D anisotropic Gaussian function was fitted to the fluorescence distribution by non-linear least squares optimization (using Matlab function "fmincon"). The full-width-at-half-maximum (FWHM) of the fitted Gaussian was used to define the contour of the corrected expression area. This approach provided the occlusion-corrected estimation of the TeNT-P2A-GFP expression surface, reported in Supplementary Fig. 12f.

### Anatomical tracing analysis

For each brain, M2 AChR immunostaining and tracer fluorescence images of each slice were realigned by matching the radial blood vessel pattern. To do so, we applied the "Normalize local contrast" function in ImageJ and then inverted the image to crop low intensity pixels and isolate blood vessel holes. We then iteratively registered these images across the stack in ImageJ using the "MultiStackReg" function with affine transformation. The transformation could then be applied to the original stack. Slices with good fluorescence signals were then averaged together. For anterograde tracing data, we first applied an "Unsharp mask" function (radius: 20, mask weight: 0.8) and pixel values were log-transformed to enhance axonal projections with respect to the strong signal at the injection site where cell bodies are labeled. For retrograde tracing data, a top hat filter (radius: 12) was applied to highlight the signal carried by CTB-positive cells followed by the "Enhance contrast" function (Saturated pixels: 0.55%). Realigned average slices for each mouse were then manually fitted to a projection of the Allen mouse brain atlas using M2 AChR landmarks[69]. Individual mouse atlases were then registered to common reference atlas. Centroid positions for injection locations were determined by calculating the center of mass of the fluorescence signals within manually outlined regions surrounding the injection sites. For anterograde tracing data, fluorescence images corresponding to different injections were normalized to the unit range per channel across brains and median-equalized across channels within brains. For retrograde tracing data, fluorescence images corresponding to different injections were rescaled to have the same peak value across channels and brains. Center-of-mass coordinates for each injection in each brain were projected on a line corresponding to the vertical meridian in the somatotopic or retinotopic map at the targeted site. This yielded, for each injection, a vertical coordinate as the center location. Lastly, we obtained an average projection map by weighting the coordinate of each injection center with its normalized pixel intensity at every pixel across the stack of fluorescence images (Fig. 4b, c, d and Supplementary Fig. 8g). The maps obtained in this way represent for every pixel the preferred projection origin (anterograde) or projection target (retrograde) location along the vertical line, allowing for an understanding of the topographic mapping of inputs and outputs associated with various regions. We also established a statistical metric to assess the significance of projections at each location. This was accomplished by comparing the median difference between two distributions: the local distribution of normalized pixel intensities at each location and the distribution of normalized pixel intensities from a specific frontal-lateral region of the slice. The latter region was selected based on its lack of fluorescence signal, as determined by an initial visual examination of the raw images. The center of the color scale used to plot the projection maps corresponds to the median projection coordinate of the injection ensemble. Average projection maps were compared to wide-field retinotopic and somatotopic maps. This was achieved by aligning a selected portion (matching the cranial window field-of-view) of the projection maps with the shared wide-field/two-photon atlas. Subsequently, we calculated the Pearson coefficient of correlation between the two images as a measure of similarity. Pixels exhibiting non-significant projection signal were discarded for this analysis.

### Optogenetic substitution experiments

To test if the dorsal cortical area RL or AL is sufficient for cross-modal generalization, we used area-specific optogenetic activation as a substitute for visually evoked responses. We expressed Channelrhodopsin-2 in the cortex by injecting AAV1.CamKII0.4.-Cre.SV40 into the whole RL or AL area of Ai32 mice. This approach induced a widespread expression of Channelrhodopsin-2 (ChR2) fused with Enhanced Yellow Fluorescent Protein (eYFP). ChR2, a light-gated cation-selective channel, generates depolarizing photo-currents in response to blue illumination, thereby allowing for the excitation of cortical neurons[74]. To illuminate the cortical surface of head-fixed mice during behavioral tasks, we built a behavioral rig equipped with a Digital Light Processing (DLP) module (STAR 3.0 EVM Monochrome LED Projection Modules, DLP6500, ViALUX GmbH). This module features a 1024 × 768 Texas Instruments micro-mirror chip and a 460 nm blue LED, integrated with an inverted tandem-lens configuration macroscope. The optical tower assembly comprised two lenses configured in an L-shaped pathway, separated by a dichroic mirror. Light from the macroscope enters through the projection lens (L1; 150 mm smc Pentax-A 645 lens) at the rear, reaching a dichroic mirror (640 nm cutoff wavelength; 60 × 60 mm; Semrock) positioned at a 45° angle. This mirror reflects the light towards the front end of the objective lens (L2; 50 mm Nikon NIKKOR lens), which projects the image onto the mouse's cortex. Stimulation intensity was adjusted, for each mouse, to ensure detection while minimizing heat dissipation in the cortex: this was achieved by operating the stimulation LED within a power range that produced an irradiance between 3.5 mW/mm² and 11.5 mW/mm². The refresh rate was set at 100 Hz with a projected pixel size measuring approximately 6.5 μm. To prevent mice from directly perceiving the blue light used for optogenetic stimulations, we enclosed the optical column and the mouse skull with an opaque sleeve. To optogenetically stimulate the cortex with task relevant information concerning the vertical position of the stimulus, we shaped the blue light projection in every trial to match the top or bottom encoding subregions of the RL or AL map. Light patterns were obtained by thresholding the average retinotopy movies within each area boundaries as defined by the fitted atlas. To effectively drive neuronal activity with blue light, we pulsed illumination at 30 Hz[74]. We used a pulsing pattern over the top or bottom encoding part lasting one second within the response window. Real-time projection control was achieved integrating ALP Vialux API with the custom-made Matlab graphical user interface (GUI) used to run the behavioral tasks and monitor mouse performance. Analysis of task performance during optogenetic substitution sessions was carried out exactly as described in the "behavioral analysis" section above.

### Network model

We constructed four recurrent excitatory (E) / inhibitory (I) rate networks representative of the areas V1, S1, AL, and RL (Fig. 8a). We abstracted the decision process to a pair of readout units which in turn drove an integrator through a drift-diffusion process. Recurrent rates of the units in each region evolved according to

$$\tau_i \dot{r}_i = -r_i + f\left(W_{ij} r_j + I(t) + \sigma_r x_i(t)\right)$$

where $W_{ij}$ are the connection weights from unit $j$ to unit $i$, $I(t)$ is a source of external input to V1 and S1 cells only, and $x_i$ is noise independent to each unit, evolving according to $dx_i = -x_i dt + \sigma_x dW$ where $dW$ is a Wiener process with scaling factor $\sigma_x$. For I cells $\sigma_r = 0$. We used a saturating transfer function, given by:

$$f(x) = \beta \left\lfloor \frac{1}{1 + \exp(-(x - x_0)/\rho)} - \frac{1}{1 + \exp(x_0/\rho)} \right\rfloor_+ .$$

where $\lfloor \cdot \rfloor_+$ denotes a threshold at zero. Each region was divided into two subnetworks (corresponding to top- and bottom-preferring cells) of $N_E$ excitatory neurons and $N_I = N_E$ inhibitory neurons each. These subnetworks were anatomically segregated such that weights were partitioned into top- and bottom-preferring units which did not interact (i.e. $W_{ij} = 0$ for $i$ top and $j$ bottom, or $i$ bottom and $j$ top; see Fig. 8). Within a subnetwork, neurons were all-to-all connected. Projections between regions respected subnetwork boundaries and were limited to E neurons; feedforward projections contacted both E and I neurons; feedback projections only contacted E neurons. Weights of feedforward projection connections (i.e. V1 or S1 to RL or AL) were scaled up relative to the within-region weights ($3.2W_{EE}$), and feedback projections (i.e. AL or RL to V1 or S1) were scaled down relative to within-region weights (RL: $0.4W_{EE}$; AL: $0.04W_{EE}$). In order to model AL/RL silencing we removed all incoming and outgoing connections to the AL/RL units. AL did not project to the decision units. Simulations were subdivided into multiple trials of length 400 ms, defined as a period of no external stimulation $t_{isi} = 200$ ms, i.e. $I(t) = 0$, together with presentation of a stimulus $I(t) > 0$ to the V1 or S1 network (mimicking the visual or tactile stimuli, respectively) for time $t_{stim} = 200$ ms which was exclusive to either top- or bottom-preferring units in one of the two sensory regions (V1 or S1). The stimulus was modeled as a step function, i.e. $I(t) = c_s$, where $c_s$ is a constant. The stimulus was randomized for each trial. In order to model rewards in our framework we considered two abstract readout units, $z_1$ and $z_2$, corresponding to 'go' and 'no-go' responses, respectively. The activity of these units was a weighted linear sum of activity from E neurons within the recurrent network $z_i(t) = \alpha W_{ij} r_j$ where $\alpha$ is a scaling constant. The network's decision was determined by producing a decision variable $\delta_i$ for each $z_i$ by integrating $z_i$ from stimulus onset in the presence of independent noise,

$$d\delta_i = z_i dt + \sigma dW$$

where $dW$ is a Wiener process with scaling factor $\sigma$. Decisions were instantiated once a fixed threshold $\theta_0$ was reached by either $z_1$ or $z_2$ resulting in the specific decision encoded by that readout unit; if neither $z_1$ nor $z_2$ crossed the threshold before the end of the stimulus period, the action was considered 'no-go'. Once a decision was reached the response $z_i$ was modified according to $\tilde{z}_i(t) = z_i(t) + c_r$, until the end of the stimulus interval. If the decision was correct relative to the presented stimulus then $c_r > 0$, otherwise $c_r < 0$. $|c_r|$ was defined as the difference between the current $z_i$ and the last value of $z_i$ before the stimulus period. Only projection weights from regions V1, S1, and RL to the output variables $z_i$ were plastic. Weights evolved according to a Hebbian rule:

$$\dot{W}_{ij} = \gamma z_i r_j$$

where $i \in \{1, 2\}$, $j \in \{1, \ldots, N_E\}$, and $\gamma$ is the learning rate. Weights were bounded between 0 and $W_{max} = 1$ and normalized after each update to the initialized mean $\bar{W}_z$ within a region. Weights were initialized as a gaussian centered at mean $\bar{W}_z = 0.5$ with standard deviation $\sigma_z = 0.1$. All model simulations were run for 100 trials before and after a switch. In order to compare model performance

with the animal behavior we recorded the trial outcome for each stimulus presentation. We then computed a performance metric and 'go' probabilities conditioned on the stimulus in a sliding window of width 15 trials. In particular, if we define $s_1$ as input to the bottom-preferring units and $s_2$ as input to the top-preferring units, within each window we calculated $P(z_1 | s_1)$ and $P(z_1 | s_2)$ as the 'go' probabilities. The performance accuracy was computed as P(correct) with correct defined as $z_1 > z_2 | s_1$ and $z_2 > z_1 | s_2$ in the rule-preserving context and $z_1 > z_2 | s_2$ and $z_2 > z_1 | s_1$ in the rule-reversing context. We quantified the number of trials to reach criterion across four different trial conditions: rule-preserving and rule-reversing switches in the full model; rule-preserving and rule-reversing switches with RL silenced. We set a criterion value of 0.75 performance accuracy for each trial condition to quantify the number of trials until the model relearned after a switch. Because the mean performance accuracy increased approximately monotonically while response conditions were held constant, we computed the first time at which a given model run reached the criterion value as the time to (re)learn. If a model run did not reach the criterion within a specified number of trials, we omitted it from this analysis. The parameters used in the model are the following:

| Variable | Value |
|---|---|
| $W_{EE}$ | 0.05 |
| $W_{EI}$ | 0.065 |
| $W_{IE}$ | 0.055 |
| $W_{II}$ | 0.045 |
| $c_s$ | 20 |
| $\rho$ | 15 |
| $x_0$ | 20 |
| $\beta$ | 30 |
| $\alpha$ | $1/(6N_E)$ |
| $\alpha$ (lesion) | $1/(2N_E)$ |
| $\gamma$ | 4e-7 |
| $\sigma$ | 0.08 |
| $\sigma_x$ | $\sqrt{2}$ |
| $\sigma_r$ | 0.75 |
| $\tau_E$ | 20 |
| $\tau_I$ | 10 |
| $\theta_0$ | 80 |

The code is available at: https://github.com/comp-neural-circuits/Guyoton-Matteucci-etal-2025.

### Quantification and statistical analysis

Statistical details of experiments and analysis are described in figure legends and in the main text. Details include statistical tests used, sample type and size as well as definition of bar plots and error bars. In figure legends, standard error of the mean (S.E.M.) is specified when plotted as error bars. Paired or unpaired $t$-tests were used to assess significance of mean comparisons (implemented by Matlab functions "ttest" and "ttest2", respectively). Normality tests were not performed systematically but individual data points were plotted to visualize distributions. Wilcoxon signed rank test was used to assess significance in paired median comparisons (implemented by Matlab function "signrank"). Wilcoxon rank sum test was used for unpaired median comparisons (implemented by Matlab

function "ranksum"). Pearson coefficient of correlation was used to compute correlations between two conditions (implemented by Matlab function "corr"). Across all fits reported uncertainties (i.e. confidence intervals) for best-fit parameter values were extracted from fit covariance matrices (fitting was performed using Matlab function "nlinfit"). In order to assess the statistical significance of multisensory modulation at each time bin, for each visuo-tactile stimulus condition, and for each responsive neuron, we performed a bootstrap-t procedure[75] with 2000 outer resamplings and 25 inner resamplings (over trials) in order to obtain $p$ values for the difference from zero of MI values.

### Reporting summary

Further information on research design is available in the Nature Portfolio Reporting Summary linked to this article.

## Data availability

The dataset used in this study is freely accessible on Zenodo at https://doi.org/10.5281/zenodo.14712478.

## Code availability

The Matlab code used in this study is freely accessible on Zenodo at https://doi.org/10.5281/zenodo.14712478. The Python code for the simulations is freely available on GitHub at https://github.com/comp-neural-circuits/Guyoton-Matteucci-etal-2025.git.

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

## Acknowledgements

We thank Denis Jabaudon, Christian Lüscher, Alan Carleton, Daniel Huber, Nader Nikbakht, Davide Zoccolan and members of the El-Boustani laboratory for discussions. This work was supported by the Swiss National Science Foundation: PCEFP3_181070 (S.E.-B.).

## Author contributions

M.G., G.M. and S.E.-B. conceived experiments. M.G. performed surgeries, viral infections, wide-field and two-photon imaging, behavioral experiments, and post-mortem analysis. M.G., G.M. and C.G.F. performed behavioral experiments with optogenetics. G.M. coded acquisition software and performed data analysis. M.P.G. and J.G. conceived the network model and performed simulations. M.G., G.M. and S.E.-B. wrote the manuscript.

## Competing interests

The authors declare no competing interests.

## Additional information

**Peer review information** : *Nature Communications* thanks Umberto Olcese, Aleena Garner and the other, anonymous, reviewers for their contribution to the peer review of this work. A peer review file is available.

