## [Transparent Peer Review file · Nature Communications]

Cortical circuits for cross-modal generalization

Corresponding Author: Professor Sami El-Boustani

Version 0:

Reviewer comments:

Reviewer #1

(Remarks to the Author)

The manuscript by Guyoton and colleagues represents a tour de force aiming to address the role of different association areas in the mouse neocortex in the process of generalizing a task across the visual and tactile modality. The manuscript is generally well done, includes an impressive amount of state-of-the-art techniques, and addresses an interesting and important question in the context of understanding what role association cortices such as RL and AL play in processing and acquiring perceptual tasks. Nevertheless, I have some major concerns that the authors should address about the differential role of RL and AL. Specifically, I believe that, in the absence of additional data supporting the different role played by RL and AL, the results should be interpreted with caution, for what pertains the association with RL (and dorsal stream areas) - but not of AL and ventral stream areas - with the process of cross-modal generalization.

MAJOR

1) One of the major claims of the authors is that inactivation of dorsal (RL) but not ventral (AL) areas impairs cross-modal generalization. First of all, I would advise the authors to use caution when generalizing the effects observed for RL and AL to other association cortices. More importantly, I believe that the authors' claim about the differential role for RL and AL should be more carefully evaluated. In particular, while inactivation of RL has a clear effect, it seems that inactivation of AL also plays a role, albeit somewhat weaker. When looking at Fig. 5j, inactivation of AL transiently reduces cross-modal generalization, even if performance after the switch does not drop to chance level like in the case of RL inactivation. Thus, the authors should carefully rephrase their claims about the differential role of RL and AL, and, moreover, also carefully assess the possible factors underlying this difference. In fact, besides intrinsic differences between RL and AL, different viral expression patterns could perhaps also explain the different results. From the presented data, it seems that RL injections generally seem to extend to V1 and S1 more than injections in AL (see also extended Figure 11). I urge the authors to carefully review the assessment of the differential impact of RL and AL, and also consider possible alternative explanations for the observed results. A detailed assessment of differential expression patterns might be relevant.

2) In view of the fact that AL also seems to play a role in cross-modal generalization, I believe that the authors should consider to revise the results presented in Fig. 6 (gain of function). I understand that repeating the experiments shown in Fig. 6 for AL would require a major effort, but this would provide some key elements to determine whether AL is also sufficient to induce cross-modal generalization. If the different role of RL vs. AL is confirmed by the gain-of-function experiment, it would significantly strengthen the claim of the authors. However, in view of the major effort required to perform these additional experiments, the authors should at least discuss the limitation presented by the lack of this data about AL.

3) I wonder to what extent the model presented in Fig. 7 would differ for AL. In other words, can the authors explain not only the potential role of RL but also the absent (or more limited) role of AL using a modeling approach? I believe that this would significantly strengthen the authors' claim. This would be very relevant in view of the fact that RL and AL show very similar properties for what pertains neural correlates of sensory processing, but a different role in the generalization process. A computational modeling approach would be very useful in addressing this important question.

MINOR

1) The authors placed a large volume of data in the extended data. I would recommend to take a careful look at the extended data and, whenever possible, move it to the main figures for ease of access and to improve the readability of the manuscript.

2) On page 45, after "Spectra Physics", the word "which" should be replaced with "whose"

(Remarks on code availability)

Reviewer #2

(Remarks to the Author)

The authors conceived and executed an excellent study addressing cross-sensory generalization. A lot of time, effort, and care clearly went into this work. Overall the manuscript is clear and well written. Very clever, very awesome paradigm to measure transfer learning between somatosensation and vision - directionality with rule preservation and rule switching. The reported spatial overlap and topographic congruence of tactile and visual areas in mouse cortex is a very important, significant finding. This result aids in filling a major gap in the field in understanding HOW different sensory modalities interact in the cortex. The functional anatomy supports the behavioral data, which sheds light on WHY, and under what circumstances, different sensory modalities interact in the cortex. The authors discovered that RL is required for tactile-visual transfer learning/ generalization to occur, but RL does not appear to affect unisensory learning. The conclusions discussed are reasonable given the results. I fully support publication of this article and believe it will substantially move the field forward. I have one major concern and a few minor points to be addressed. Please see below.

Major:

RL is necessary for animals to perform at the same level after modality switching (e.g. tactile to visual). However, it is not clear why mice do not perform well when the contingency is switched (e.g. top ->bottom). Acquiring single cell, 2P data during the task for the matched and switched conditions would be one way of describing functional differences between these two conditions and animals' ability to perform in them (e.g. when the mouse performs correctly in the tactile task, do the same neurons respond in RL during the visual task even when the contingency is switched (top->bottom) or does the activity in RL just degrade or become uncorrelated in some general sense)? This is also important because the task requires engagement and reinforcement, which changes the dynamics of sensory responses quite a bit. Perhaps this concern can also be addressed using the neural network model (e.g. the connectivity from 'V1' to 'RL' to 'S1' is stronger for congruent units than incongruent units).

Minor:

In reference to into line, "While neuronal correlates of increasingly abstract representations have been reported 6–9," please briefly state a couple of specific examples or an overarching example of what you mean by "abstract" and add locations (e.g. in higher association cortices). This will provide readers with a more concrete foundation on which to consider the open question you mention next.

In the discussion, please mention how cross-modal generalizations may or may not occur between other sensory modalities with different anatomical connections than the visual-tactile network. In other words, are visual-tactile multisensory interactions special or have you described a general mechanism for any two sensory modalities?

The finding that mice rapidly generalize the rule from a whisker tactile task to a visual task is remarkable and very significant. I think this result will surprise many researchers in the field of vision, and will be the basis of more visual studies using mice.

Do mice learn the rule easily because visual flow and whisker stimulation naturally correspond, for example as mice move about their home cages? Would mice raised in the dark and exposed to light only under controlled conditions in which visual and tactile information never match (e.g. visual flow on a monitor never corresponds to directionality or speed of whisker stimulation) show the same rapid generalization learning? These experiments are probably beyond the scope of this project but may be worth mentioning in the discussion.

Line 96: parenthesis around 'green' to match style of other colors and their definitions and to make the color easier to find in the text.

Fig 1 d. I assume the drop in blue just before trial 500 is because mice become satiated during the task. In order to evaluate the data this plot needs to show initial performance for the tactile learning (early session- presumably at chance?) Also, by computing the traces on a sliding window of 60 trials, behavior responses on early trials right after the switch occurs is lost. This is important to show to reflect how quickly mice actually transferred the rule. It looks like mice licked non-specifically to both visual conditions for the first few trials and then, after not being rewarded for one, reduced licking for that condition. Please show this trial-by-trial data.

Fig 1 e. Again, the plot does not reflect any learning. In order to understand the data, we need to see when the mice improved from chance to ~80%.

Fig 1. legend has organization typos.- for example, the second (d) at line 100 looks like it belongs with the far right bar plot.

I think the bidirectionality shown in extended data figure 1g is very important and should be included in the main figure 1.

1h is a super cool result.

Regarding results mentioned in line 173-176, please make clear in methods how many trials and sessions lead to this result (e.g. avg # +/- SEM) as the amount of training likely plays a role in animals' flexibility to unlearn and /or learn a new rule or rule switch.

Line 182 "necessary for conscious perception" while true, this isn't relevant for the paper. If you want to address this point, you'd need to somehow train mice unconsciously (like move a whisker and stimulate the VTA in anesthetised animals) and then wake them up and train them on the visual task. Otherwise, I don't think conscious perception can be claimed. Please rephrase this in your text?

Fig 2. Beautiful maps and nice demonstration of spatially overlapping modalities.

In fig. 4 the number of neurons is stated. Please also indicate the number of mice used in these experiments. 4b is an interesting result- in how many mice did you observe this? Are data in this fig from the same 25 mice represented in Ext. Data Fig. 10?

Also, Ext. Data Fig 10 d. Please show or state any (statistical) differences between neuron types for the decoder instead of just within neuron types comparing tactile decoding and visual decoding. This will make your point stronger.

It is not clear what the visuo-tactile sparse noise stimulus is. Please reference the main text with the description in the methods section. In the methods, it looks like the visual portion is a filtered sparse noise stimulus, but what is the tactile stimulus? What area of space is covered? Are stimuli presented congruently or distinctly or both and for what fraction of presentation time? This is important for understanding what the Ca2+ imaging data reflects. Please add a more detailed, clearer description to the methods.

Line 429: "Mice normally learned" this was confusing. Would be clearer to write, Mice learned the whisker discrimination task at the same rate as control mice- if this is what you mean.

Fig 5c. Why did these mice satiate in about half the time as the non TeNT mice (red line drops to chance by trial ~ 250)? Please explain.

532 What do you mean by "after mice began responding to optogenetic blue light stimulations" Please state explicitly. Why did mice not pick up this pattern immediately as they did in the visual stimulus condition (as shown in fig 1 e)? This discrepancy needs to be addressed. The result is still awesome, but if the optogenetic learning rate doesn't match the visual learning rate, then it cannot really be said that the optogenetic stimulation is sufficient for visuo-tactile generalization.

The neural network model fits this data well, but it is not clear to me if the model is informative in a more general sense. What do we learn from it that we did not learn from the data? How can it be applied to form a hypothesis that was not possible using the data alone? Please give a bit more information in this regard.

(Remarks on code availability)

I do not believe the code is available yet. It looks like it will be upon publication. The authors' state "The complete data set and Matlab analysis code will be made freely available at the CERN database, Zenodo." This may or may not be a usable resource for the community and the inclusion of instructions for how to run the code should be addressed. Also, it is helpful to have an actual link. It looks like the code for the model will also be available upon publication here <https://github.com/comp-neural-circuits> but until it is, I cannot assess its usability.

Reviewer #3

(Remarks to the Author)

Guyoton, Matteucci and collaborators (GM) study cross-modal (tactile, visual) generalization. Mice were trained to discriminate (go/no-go) between top vs. bottom tactile (T) whisker stimuli. They were then tested on discrimination between visual stimuli presented in the top vs. bottom of the visual (V) field. Animals performed the visual task without additional training, only when the association of go/no-go with top/bottom was preserved. Generalization in the reverse direction was also studied.

To study the mechanisms of cross-modal generalization, GM performed functional mapping, anatomical tracing, and optogenetic stimulation (LoF, GoF) experiments. Those experiments support the conclusion that associative area RL, where neurons exhibit marked multisensory enhancement of their responses, has a causal role in mediating cross-modal generalization.

Overall GM studied an interesting problem and used a host of powerful experimental techniques which substantially advance our understanding of how the brain performs cross-modal generalization. However, the presentation of multiple facets of the data is oversimplified and not entirely consistent with the actual findings. The subtleties of the existing data that are overlooked in the text could be important, and should be explored further based on the computational model.

Major points:

GM's main finding, that RL mediates cross-modal generalization, is well supported by the experiments and controls. However, the description of the behavioral experiments overstates the symmetry between V-to-T generalization and T-to-V generalization. Inspection of Extended Data Fig.1 a-c vs. j-l suggests that behavioral generalization dynamics of V→T vs. T→V are in fact different. GM neglect to mention those differences and thus appear to be attributing them to random variability. Importantly, the anatomical tracing and functional mapping experiments also show marked asymmetries between visual and tactile processing.

As examples, notice that:

- a. Multisensory enhancement is rather strong in S1 (similar in magnitude to RL, Fig. 2f), while V1 shows negligible multisensory enhancement.
- b. Retrograde tracing from S1, V1 shows that S1 receives significant input from V1 (this result is inconsistent with the current discussion, line 651); while the reverse is not true (Fig. 3d vs. Extended Fig. 7g). Moreover, despite having done retrograde tracing from area RL (Extended Fig. 7h), GM do not show summary data for those experiments. Such data could support or refute the notion that RL integrates somatosensory and visual information "on equal footing".

Thus, multiple facets of GM's data (behavior, functional mapping, anatomical tracing) are inconsistent with symmetric integration of V and T stimuli leading to multisensory generalization. My opinion is that, given the substantial amount of experimental work that has already been performed in this study, experimentally exploring those asymmetries further goes beyond the scope of the paper. However, the computational network model in the current version – which is defined to be completely symmetric – is not consistent with the empirical findings.

In summary, (a) The aforementioned asymmetries should be discussed explicitly; (b) The projections from S1 and V1 to RL should be quantified based on the retrograde tracing, or at least GM should explain why such quantification is not feasible; and (c) The model should be used to test whether those asymmetries (anatomical/functional/behavioral) are consistent with each other.

Minor points:

From the text, it is not clear what the word "learning" refers to. In the context of the results in the paper, it could refer to learning to associate top/bottom with go/no-go, or learning to generalize visual with tactile stimuli. GM should use more precise language emphasizing they refer to the former.

GM refer to the task as involving "conscious perception". However, no experiments were done to suggest that consciousness has anything to do with the cross-modal generalization studied in the current manuscript. For example, it could be that the neural activity mediating this generalization is also present during sleep. I believe the word conscious should be removed.

In Fig. 1e,g, Fig. 6c,d (and associated Extended Data Figures), the points corresponding to sessions -1 and 1 are connected. This is confusing, because it makes it seem as if the change in lick probability occurs before the modality switch. Those points should not be connected or connected with a dashed line.

In Extended Data Fig. 8g,h, the fields of view do not span the entirety of the cranial window. Areas not within a field of view should be indicated by a different color from areas that are covered yet have low density of imaged/responsive neurons. Currently one cannot tell if dark blue means no coverage or low density.

The methods section and Extended Data Figures do not make it clear enough what percentage of neurons in area RL are affected by the optogenetic manipulation or what their response profile looks like. Providing those data will strengthen the validity of the experiments.

As GM say, the results in Fig. 5e,f suggest that area RL has a unique role in multisensory generalization, compared for example with area AL. However the description of Fig. 5h,j overstates the degree to which this is confirmed by the silencing experiments. Indeed, comparing Fig. 5h to Fig. 5j shows rather small differences in effect size, even though the spatial extent of silencing in RL is larger than in AL (compare Fig. 5g,i). The authors should tone down the text describing those results, leaving the door open to the possibility that AL also plays an important role in this computation.

In lines 662-666, GM discuss how surround suppression of incongruent stimuli may explain multisensory enhancement. Since multisensory enhancement only occurs when stimuli are congruent, wouldn't surround suppression explain multisensory suppression and not enhancement?

Tactile stimuli are the movement of either B2 or C2 whiskers, while preferred whisker position and tuning are reported as a continuous measurement. The methods section does not include an explanation of how the preferred position was computed.

Since each session has a different number of trials, it would be helpful to show in the Extended Data Figures (e.g., Extended Data Fig. 1b,c,e,f,h,i,k,l) the range of number of trials in each session. This would be important if, for example, the number of trials after a switch is different from the typical number of trials in other sessions.

I could not find information describing how session length was determined, though from Fig. 1 it is evident that the number of trials in each session was varied. Moreover, what was the time interval between sessions? This is important since there is a drop in lick rate in the last trials of the sessions shown in Fig. 1d,f. Does this arise from quenched thirst? If so, does the interval between sessions ensure similar thirst levels at the beginning of each session?

(Remarks on code availability)

Reviewer #4

(Remarks to the Author)

(Remarks on code availability)

Version 1:

Reviewer comments:

Reviewer #1

(Remarks to the Author)

The authors have addressed most of my comments. In particular the updated model (Fig. 8) and the added gain-of-function experiments focused on AL (fig. 7) clarify some key aspects. Nevertheless, I was not fully satisfied by the updated loss-of-function experiments (Fig. 8). I believe that the authors should further improve this part of the manuscript. Below I outline my remaining concerns:

1) The authors' claim that inactivation of area RL but not AL impairs performance after modality switch is based on the finding that inactivation of RL (but not AL) induces chance performance. However, this does not seem to be quantified in Fig. 6h,j. The analysis is mentioned in the legend and main text, but the lack of asterisks in the barplots indicates no difference from chance. However, I assume this was a mistake. I also wonder if the authors might compare performance after RL vs AL inactivation. Although not done in a paired setting, this comparison would further strengthen their claims.

2) The authors suggest that the reduced performance observed following AL inactivation might stem from an expression pattern in deep layers extending to AL. However, a more straightforward explanation might simply be that AL also plays a similar - albeit weaker - than RL. I recommend that the authors expand the discussion of possible confounds and alternative explanation, and give them a more prominent role in the Discussion (now this part is in the Results section, at lines 569-572).

3) The authors mention that there was no correlation between staining in V1 and S1 and a behavioral effect, but this is not shown. The authors should provide such data.

4) Overall, in the rebuttal the authors mention that the injection sites are not specifically confined to AL and RL. Correctly, they talk of a ventral (LI/AL) and a dorsal (RL/A) site. However, this careful description is not evident in the discussion. In order to make the manuscript better comparable to other studies, in which areas such as A and AM (but also RL) are often referred to as posterior parietal cortex, I would recommend that the authors revise their discussion to more carefully indicate which results specifically hold for RL and AL, and which should instead be assigned to broader subdivision of the mouse posterior association cortex.

(Remarks on code availability)

Reviewer #2

(Remarks to the Author)

The authors have sufficiently addressed all concerns, and I highly recommend this manuscript for publication. This research on cross-modal learning and generalization is impactful and extremely valuable to our understanding of mechanisms of cognition.

(Remarks on code availability)

I have reviewed the matlab code, which is clear and is a usable resource for the community.

Reviewer #3

(Remarks to the Author)

The revised version adequately addresses my comments. Please see below a few remaining minor points:

The results of the new gain of functions experiments in AL (i.e. AL activation contrary to RL is not sufficient for generalization), together with the overlap correlation analyses, justify the authors emphasis on RL's contribution to cross-modal generalization. I still believe however that the authors can be a little more cautious. Perhaps describe RL's role as "major", "outsized", "dominant", instead of "unique".

The differential levels of task difficulty for visual and tactile modalities could potentially explain the different levels of generalization in the $T \rightarrow V$ and $V \rightarrow T$ conditions. However, that still does not rule out the potential contribution of the asymmetry in the connections between S1 and V1. Relatedly, I find it strange that the model shows no functional asymmetry for any of the structural asymmetry values explored (light blue lines in Extended Data Fig. 14e,f). Here too I think the authors should be a bit more cautious and mention the possibility that asymmetry in connectivity contributes to asymmetry in generalization, especially since those asymmetries are "in the same direction".

When reading the revised manuscript, I noticed that the definition of "detection performance" is rather difficult to make sense of. The readability of the paper will benefit if this definition is unpacked.

(Remarks on code availability)

Reviewer #4

(Remarks to the Author)

(Remarks on code availability)

Point-by-point response to the reviewers' comments

Reviewer #1 (Remarks to the Author):

The manuscript by Guyoton and colleagues represents a tour de force aiming to address the role of different association areas in the mouse neocortex in the process of generalizing a task across the visual and tactile modality. The manuscript is generally well done, includes an impressive amount of state-of-the-art techniques, and addresses an interesting and important question in the context of understanding what role association cortices such as RL and AL play in processing and acquiring perceptual tasks. Nevertheless, I have some major concerns that the authors should address about the differential role of RL and AL. Specifically, I believe that, in the absence of additional data supporting the different role played by RL and AL, the results should be interpreted with caution, for what pertains the association with RL (and dorsal stream areas) - but not of AL and ventral stream areas - with the process of cross-modal generalization.

We thank the reviewer for their insightful and constructive comments. To address the concerns raised, we have conducted additional optogenetic experiments, model simulations, and performed new analyses. These efforts are detailed in the revised manuscript. We believe the new results provide substantial support for our conclusions regarding the distinct roles of RL and AL in cross-modal generalization.

MAJOR

1) One of the major claims of the authors is that inactivation of dorsal (RL) but not ventral (AL) areas impairs cross-modal generalization. First of all, I would advise the authors to use caution when generalizing the effects observed for RL and AL to other association cortices. More importantly, I believe that the authors' claim about the differential role for RL and AL should be more carefully evaluated. In particular, while inactivation of RL has a clear effect, it seems that inactivation of AL also plays a role, albeit somewhat weaker. When looking at Fig. 5j, inactivation of AL transiently reduces cross-modal generalization, even if performance after the switch does not drop to chance level like in the case of RL inactivation. Thus, the authors should carefully rephrase their claims about the differential role of RL and AL, and, moreover, also carefully assess the possible factors underlying this difference. In fact, besides intrinsic differences between RL and AL, different viral expression patterns could perhaps also explain the different results. From the presented data, it seems that RL injections generally seem to extend to V1 and S1 more than injections in AL (see also extended Figure 11). I urge the authors to carefully review the assessment of the differential impact of RL and AL, and also consider possible alternative explanations for the observed results. A detailed assessment of differential expression patterns might be relevant.

We thank the reviewer for their detailed and thoughtful feedback. We have addressed the concerns regarding the differential roles of RL and AL as follows.

First, we have clarified in the revised manuscript that many of our findings do not refer to specific areas in isolation but rather to clusters of ventral and dorsal areas. The hotspots identified through wide-field and two-photon recordings, as well as TeNT injections, overlap with RL/A and LI/AL. However, for certain additional experiments and analyses, we focused

explicitly on the distinction between RL and AL as they predominantly contain visuo-tactile neurons of the dorsal and ventral cluster, respectively.

Our analysis based on TeNT-P2A-GFP, which specifically selected expression covering RL and AL, revealed a complete lack of generalization for RL inactivation and a transient reduction in performance for AL. It is important to note that this classification did not initially account for variability in injection site and spread. To address this, we performed additional analysis in the original manuscript, demonstrating that only GFP overlap with area RL shows correlates with impairment in generalization (Figure 6e-f of the revised manuscript). We emphasized the strength of these analysis in the revised manuscript at page 22, lines 516-519.

We acknowledge the reviewer's observation that AL inactivation slightly impairs performance after modality switch. A possible explanation is imprecision in estimating the expression pattern. Specifically, deeper-layer spread of expression (see image below), undetectable from the surface, could sometimes happen and affect nearby areas without being identified through wide-field imaging. This limitation, along with the limitations of using GFP expression in wide-field imaging to define the inactivated circuit, is now explicitly discussed in the revised manuscript at page 24, lines 567-572.

Figure. An example of a coronal brain slice showing the spread of TeNT-P2A-GFP injection within deep layers to nearby areas.

Regarding TeNT-P2A-GFP expression patterns, our previous analysis did not take into account the masking effect caused by the edge of the cranial window which could lead to an underestimation of the expression spread for ventral areas that are very lateral and therefore often close to the edge. Indeed, all injections were carried out with the same parameters (injected volume, depth, etc.), expected to result in a comparable expression profile. To address this, we performed an additional analysis comparing the extent of expression patterns for injections in ventral and dorsal areas. Using a Gaussian fit, we estimated the actual size of expression patterns, including regions obstructed by the cranial window. This analysis indicates that the expression sizes for ventral and dorsal injections are comparable. This is now presented in Extended Data Figure 12f and mentioned in the main text at page 22, lines 513-516.

Finally, we acknowledge that controlling for overlap with V1 and S1 is challenging due to the different anatomical locations of RL and AL. However, our analyses show that overlap with these areas does not correlate with generalization impairment as discussed at page 22, lines 523-525.

2) In view of the fact that AL also seems to play a role in cross-modal generalization, I believe that the authors should consider to revise the results presented in Fig. 6 (gain of function). I understand that repeating the experiments shown in Fig. 6 for AL would require a major effort, but this would provide some key elements to determine whether AL is also sufficient to induce cross-modal generalization. If the different role of RL vs. AL is confirmed by the gain-of-function experiment, it would significantly strengthen the claim of the authors. However, in view of the major effort required to perform these additional experiments, the authors should at least discuss the limitation presented by the lack of this data about AL.

We thank the reviewer for suggesting these experiments. We conducted a new series of gain-of-function experiments using optogenetic activation of AL in mice expressing ChR2 in this area.

Unexpectedly, activation of AL did not evoke behavioral responses in these mice, even when we applied stronger blue light intensity or matched the pattern size to those previously used for RL activation. Following the modality switch, we trained these mice for over seven days without observing any signs of opto-triggered licking responses. When comparing these results with the time required for mice to adapt to optogenetic stimulation in other conditions, we concluded that mice were unable to generalize sensorimotor associations learned with tactile stimulation to direct activation of AL. In contrast, mice adapted quickly to optogenetic stimulation of RL when task rules were preserved and more slowly when rules were reversed.

These findings align with our original conclusions that area RL, but not AL, is sufficient to mediate cross-modal generalization. We believe that stimulations of AL fail to elicit responses, at least in part, due to its weak projections to motor regions compared to area RL (see following point). These new results are now presented in Figure 7 of the revised manuscript. Furthermore, we have improved the description of the optogenetic experiments, including a detailed analysis of habituation to optogenetic stimulation at pages 27-28, lines 635-662.

3) I wonder to what extent the model presented in Fig. 7 would differ for AL. In other words, can the authors explain not only the potential role of RL but also the absent (or more limited) role of AL using a modelling approach? I believe that this would significantly strengthen the authors' claim. This would be very relevant in view of the fact that RL and AL show very similar properties for what pertains neural correlates of sensory processing, but a different role in the generalization process. A computational modelling approach would be very useful in addressing this important question.

We thank the reviewer for raising this important point. Given the similarity between visuo-tactile neurons in the dorsal and ventral streams, as well as the organized spatial maps in these areas, it is indeed crucial to explain why these regions might contribute differently to sensorimotor learning using our model.

To address this, we incorporated previously published evidence regarding differences in connectivity patterns between these areas and other cortical regions.

In particular, Wang et al. (2012) [<https://doi.org/10.1523/JNEUROSCI.6063-11.2012>] characterized in great detail the connectivity patterns involving ventral and dorsal associative areas. Their results highlight two critical aspects that are relevant to our modeling approach:

- 1) In line with the primate literature, ventral areas such as AL have significantly fewer projections to motor cortices compared to dorsal areas. Specifically, RL exhibits much stronger projections to the pre-motor cortex (M2) than AL.
- 2) Feedback projections from AL to primary sensory cortices (V1 and S1) are much weaker compared to those originating from RL.

Together, these findings highlight an important contrast between AL and RL that could explain the differential contributions of these areas to cross-modal generalization. The weaker projections from AL to the pre-motor cortex likely prevent direct activation of AL from eliciting a behavioral response, while the lack of strong feedback projections from AL to primary sensory cortices limits the reverberation necessary for generalization. This provides insight into why AL inactivation does not impair cross-modal generalization and why direct activation of AL is insufficient to generalize the task.

To address this further, we added new simulations incorporating AL into our circuit model, using connectivity patterns consistent with the published findings mentioned above. These simulations show that a circuit with such connectivity differences can account for the distinct roles of RL and AL in cross-modal generalization. The results of these new simulations are now presented in Figure 8 and Extended Data Figure 14 of the revised manuscript (pages 28-31), strengthening our conclusions.

MINOR

- 1) The authors placed a large volume of data in the extended data. I would recommend to take a careful look at the extended data and, whenever possible, move it to the main figures for ease of access and to improve the readability of the manuscript.

Following the reviewer's comment, we have moved the results previously presented in Extended Figure 1 to a main figure, which now appears as Figure 2 in the revised manuscript.

- 2) On page 45, after "Spectra Physics", the word "which" should be replaced with "whose"
Ok.

Reviewer #2 (Remarks to the Author):

The authors conceived and executed an excellent study addressing cross-sensory generalization. A lot of time, effort, and care clearly went into this work. Overall the manuscript is clear and well written. Very clever, very awesome paradigm to measure transfer learning between somatosensation and vision - directionality with rule preservation and rule switching. The reported spatial overlap and topographic congruence of tactile and visual areas in mouse cortex is a very important, significant finding. This result aids in filling a major gap in the field in understanding HOW different sensory modalities interact in the cortex. The functional anatomy supports the behavioral data, which sheds light on WHY, and under what circumstances, different sensory modalities interact in the cortex. The authors discovered that RL is required for tactile-visual transfer learning/ generalization to occur, but RL does not appear to affect unisensory learning. The conclusions discussed are reasonable given the results. I fully support publication of this article and believe it will substantially

move the field forward. I have one major concern and a few minor points to be addressed. Please see below.

We thank the reviewer for their comments and assessment of our work. In response, we have conducted additional experiments, simulations, and analyses to address the concerns below. We believe that these additions have further strengthened the conclusions of the manuscript and improved its overall clarity and rigor.

Major:

RL is necessary for animals to perform at the same level after modality switching (e.g. tactile to visual). However, it is not clear why mice do not perform well when the contingency is switched (e.g. top ->bottom). Acquiring single cell, 2P data during the task for the matched and switched conditions would be one way of describing functional differences between these two conditions and animals' ability to perform in them (e.g. when the mouse performs correctly in the tactile task, do the same neurons respond in RL during the visual task even when the contingency is switched (top->bottom) or does the activity in RL just degrade or become uncorrelated in some general sense)? This is also important because the task requires engagement and reinforcement, which changes the dynamics of sensory responses quite a bit. Perhaps this concern can also be addressed using the neural network model (e.g. the connectivity from 'V1' to 'RL' to 'S1' is stronger for congruent units than incongruent units).

We appreciate the reviewer's comment and acknowledge that the description of the hypothesized circuit mechanism in the original manuscript may have lacked clarity. To address this, we emphasize that the mechanism we propose does not rely on changes in sensory neurons under the different task conditions.

Our interpretation lies in the properties of multimodal neurons in our dataset. The vast majority of multimodal neurons are tuned to spatially congruent stimuli in both modalities, while incongruently tuned neurons were found to be very rare. Due to their congruent tuning, these neurons are well-suited to support generalization.

When the task switches to a new modality with the same rule, decisional neurons downstream to sensory areas continue to rely on the same population of multimodal neurons without altering their interpretation of the signals, allowing mice to maintain high task performance. This was already demonstrated in our decoding analysis, and was further supported by our computational model, which show similar results.

Regarding the poor performance of mice during rule-reversing switches, we interpret this as successful generalization of the original spatial rule rather than a failure to generalize. Mice continued to apply the same rule after the switch, indicating that they generalized the identity of the stimuli (with the wrong rule). The drop in performance (in the data and in the model) arises due to the reversal of the rule rather than an inability to generalize. A similar phenomenon is observed when rules are reversed within the same modality, although relearning in this case is slower, likely due to habit formation (Extended Data Figure 4 in the revised manuscript).

For these reasons, we believe that performing two-photon imaging of sensory responses in RL during task performance may not be strictly necessary to explain cross-modal generalization, as the functional properties of multimodal neurons already account for this phenomenon. We do not anticipate changes in multisensory neuron responses around the task switch, as functional maps aligned across modalities are already present in naïve mice. Instead, the changes likely occur at the synaptic level between sensory areas and motor or

decisional areas, where sensory inputs are remapped to corresponding motor outputs in a goal-directed manner following a switch. We have clarified these points in the revised manuscript at page 19, lines 439-444; page 21, lines 490-492 and page 29, lines 682-689.

Minor:

In reference to into line, “While neuronal correlates of increasingly abstract representations have been reported 6–9,”, please briefly state a couple of specific examples or an overarching example of what you mean by “abstract” and add locations (e.g. in higher association cortices). This will provide readers with a more concrete foundation on which to consider the open question you mention next.

We clarified the definition of “abstract representation” and provided additional examples with their corresponding areas at page 3, lines 46-53.

In the discussion, please mention how cross-modal generalizations may or may not occur between other sensory modalities with different anatomical connections than the visual-tactile network. In other words, are visual-tactile multisensory interactions special or have you described a general mechanism for any two sensory modalities?

We have added a paragraph to the discussion addressing the broader applicability of cross-modal generalization to other sensory modalities, such as audio-visual interactions, and their corresponding circuit requirements (pages 33-34, lines 794-803).

Additionally, we performed new simulations using our model to demonstrate that specific aspects of anatomical connectivity are necessary for cross-modal generalization to occur. These include differences in feedback connectivity and projections to motor regions, which could explain varying contributions even among visuo-tactile areas with aligned functional maps. These findings are now illustrated in Figure 8 of the reviser manuscript.

The finding that mice rapidly generalize the rule from a whisker tactile task to a visual task is remarkable and very significant. I think this result will surprise many researchers in the field of vision, and will be the basis of more visual studies using mice.

Do mice learn the rule easily because visual flow and whisker stimulation naturally correspond, for example as mice move about their home cages? Would mice raised in the dark and exposed to light only under controlled conditions in which visual and tactile information never match (e.g. visual flow on a monitor never corresponds to directionality or speed of whisker stimulation) show the same rapid generalization learning? These experiments are probably beyond the scope of this project but may be worth mentioning in the discussion.

We agree with the reviewer that these are fascinating questions. Indeed, we believe that development in natural conditions is essential to stabilize these functional maps and establish the precise correspondence between visual and whisker tactile inputs. Experiments involving sensory alteration (visual, tactile, or both) would be necessary to dissect the post-natal development of these circuits during critical periods of plasticity for multisensory integration. While such experiments are beyond the scope of this study, we recognize their potential to advance our understanding of these processes and are hoping to explore this exciting avenue in future work. To address this point, we have added a paragraph to the discussion on circuit plasticity during development (pages 35-36, lines 835-853).

Line 96: parenthesis around 'green' to match style of other colors and their definitions and to make the color easier to find in the text.

Ok.

Fig 1 d. I assume the drop in blue just before trial 500 is because mice become satiated during the task. In order to evaluate the data this plot needs to show initial performance for the tactile learning (early session- presumably at chance?) Also, by computing the traces on a sliding window of 60 trials, behavior responses on early trials right after the switch occurs is lost. This is important to show to reflect how quickly mice actually transferred the rule. It looks like mice licked non-specifically to both visual conditions for the first few trials and then, after not being rewarded for one, reduced licking for that condition. Please show this trial-by-trial data.

Indeed, the drop in lick responses at the end of each session corresponds to when the mouse becomes disengaged due to satiation. These epochs are excluded from the analysis of performance.

Regarding the analysis of lick behavior and performance immediately following the switch, it is necessary to use a window of trials to assess behavioral response rates. However, as the reviewer points out, we did not present the detailed data for the first session in the original version of the manuscript. Typically, mice performed above chance level already during the first 10 trials in the rule-preserving switch, whereas they performed below chance level for a short period after the rule-reversing switch before disengaging.

These behaviors demonstrate that mice maintained the same rule when switching modalities, even during the earliest part of the session. This is illustrated in the Figure below and is now clarified on page 6-7, lines 126-129 and 135-138 of the revised manuscript.

Figure. Behavior of mice in the first trials following rule-preserving or rule-reversing modality switches. a, Conditional lick probabilities and performance averaged across mice for the last session before the modality switch. A sliding window of 10 trials is used to capture rapid changes in behavior during the early part of the session. Lick probability is shown for go stimuli (blue), no-go stimuli (red), and in the absence of stimulus (purple). Performance is indicated in green. Shaded areas represent s.e.m., and the gray area indicates the first 100 trials of the session. b, Same as panel a for the first session following a rule-preserving modality switch. c, Same as panel a for the first session following a rule-reversing modality switch. d, Distribution of lick probabilities and performance averaged over the gray areas of panels a–c. The plot shows that mice perform above chance level shortly after a rule-preserving modality switch, whereas they perform below chance level following rule-reversing switches confirming that they applied the rule learned during the first task.

Fig 1 e. Again, the plot does not reflect any learning. In order to understand the data, we need to see when the mice improved from chance to ~80%.

As the learning curves of mice for the two-whisker task were already described in a previous work (Matteucci et al, 2022, doi: 10.0.3.248/j.neuron.2022.09.032), we focused primarily on the generalization aspect in the manuscript. However, we agree that it is important to confirm that mice are able to properly learn the task across all conditions. To address this, we have added information about the learning of the first task to Extended Figure 2 of the revised manuscript, which now shows comparable learning trajectories across conditions.

Fig 1. legend has organization typos.- for example, the second (d) at line 100 looks like it belongs with the far right bar plot.

Thank you for pointing out this issue. In the revised manuscript, we have corrected the organization of the legends and ensured that the panel references are clear and consistent. Specifically, we have only kept bold letters for first references to panels to avoid any potential confusion.

I think the bidirectionality shown in extended data figure 1g is very important and should be included in the main figure 1.

Thank you for the suggestion. We agree that the bidirectionality shown in Extended Data Figure 1g is important. In the revised manuscript, we have moved some of the data previously presented in Extended Data Figure 1 to the main text, where it now appears in Figure 2.

1h is a super cool result.

Regarding results mentioned in line 173-176, please make clear in methods how many trials and sessions lead to this result (e.g. avg # +/- SEM) as the amount of training likely plays a role in animals' flexibility to unlearn and /or learn a new rule or rule switch.

To address the reviewer's concern and ensure that mice receive a comparable level of training before being exposed to modality switches, we performed a detailed analysis. In Extended Data Figure 2 of the revised manuscript, we compare the number of sessions, performance levels, weight loss, and session duration prior to modality switches across conditions. This analysis demonstrates that mice have a comparable level of experience and motivation before being exposed to any switch. These details are now discussed on page 7, lines 140-146 and page 9, lines 186-189.

Line 182 "necessary for conscious perception" while true, this isn't relevant for the paper. If you want to address this point, you'd need to somehow train mice unconsciously (like move a whisker and stimulate the VTA in anesthetised animals) and then wake them up and train them on the visual task. Otherwise, I don't think conscious perception can be claimed. Please rephrase this in your text?

We have rephrased this sentence in the revised manuscript.

Fig 2. Beautiful maps and nice demonstration of spatially overlapping modalities.

In fig. 4 the number of neurons is stated. Please also indicate the number of mice used in these experiments. 4b is an interesting result- in how many mice did you observe this? Are data in this fig from the same 25 mice represented in Ext. Data Fig. 10?

In the revised manuscript, we have added the number of mice used in this dataset.

Regarding non-linear visuo-tactile responses, we have quantified the proportion of neurons exhibiting strong non-linearity across our dataset, as illustrated in the Figure below. Additionally, we now report in what proportion of mice these responses were observed. This is now reported at page 19-20, lines 461-471.

Finally, we have explicitly stated on pages 95 and 97 that the neurons in Figure 5 and Extended Data Figure 11 of the revised manuscript are from the same dataset.

*Figure: Distributions of multimodal response nonlinearity, quantified by the norm of the difference between observed and predicted selectivity, for dorsal (magenta, n=124 neurons) and ventral (green, n=75 neurons) areas (unpaired two-sided t-test, ***p=7.6x10⁻⁴). The red dashed line indicates the threshold for strong modulation (values > 0.3), and the numbers above the red line represent the percentage of neurons with strong modulation in each population.*

Also, Ext. Data Fig 10 d. Please show or state any (statistical) differences between neuron types for the decoder instead of just within neuron types comparing tactile decoding and visual decoding. This will make your point stronger.

Ok.

It is not clear what the visuo-tactile sparse noise stimulus is. Please reference the main text with the description in the methods section. In the methods, it looks like the visual portion is a filtered sparse noise stimulus, but what is the tactile stimulus? What area of space is covered? Are stimuli presented congruently or distinctly or both and for what fraction of presentation time? This is important for understanding what the Ca²⁺ imaging data reflects. Please add a more detailed, clearer description to the methods.

We apologize for the lack of clarity in the original manuscript. To address this, we have expanded the description of the visuo-tactile sparse noise stimulus in the methods section, providing all the necessary details regarding the protocol. These updates can be found on page 51-52, lines 1204-1214 of the revised manuscript.

Line 429: "Mice normally learned" this was confusing. Would be clearer to write, Mice

learned the whisker discrimination task at the same rate as control mice- if this is what you mean.

Thank you for the suggestion. We have clarified the sentence to state, "Mice learned the whisker discrimination task at the same rate as control mice," following the reviewer's recommendation.

Fig 5c. Why did these mice satiate in about half the time as the non TeNT mice (red line drops to chance by trial ~ 250)? Please explain.

There is variability in the number of trials with engaged behavior across sessions, as shown in Figure 1d, right panel, for control mice. However, the average session duration across mice is comparable across groups, with values around 400 trials per session (N=20 mice for control group, N=25 mice for TeNT group, unpaired two-sided t-test, $p=0.89$). We have clarified this point and provided additional details in the revised manuscript at page 21, lines 505-508.

532 What do you mean by "after mice began responding to optogenetic blue light stimulations" Please state explicitly. Why did mice not pick up this pattern immediately as they did in the visual stimulus condition (as shown in fig 1 e)? This discrepancy needs to be addressed. The result is still awesome, but if the optogenetic learning rate doesn't match the visual learning rate, then it cannot really be said that the optogenetic stimulation is sufficient for visuo-tactile generalization.

We thank the reviewer for raising this point. Optogenetic stimulations of cortical areas, especially higher-order areas, are not guaranteed to evoke a perceptual experience comparable to natural vision and may have a modulatory effect (i.e. biased or distorted ongoing perception) rather than producing outright phosphenes or perception of illusory visual objects, as reported in several previous studies (see for instance Azadi et al., 2023; <https://doi.org/10.1016/j.cub.2022.12.021>). As a result, responding to optogenetic stimulations of small cortical domains might require some habituation. To address this, we first submitted mice to an optogenetic detection task to elicit consistent behavioral responses before introducing the discrimination task.

During this opto-habituation phase, we only used the rewarded stimulus (go stimulus) until we could reliably detect behavioral responses. We then introduced no-go stimuli to test generalization. Mice that underwent a rule-preserving switch with RL activation started responding to these stimuli after a day and successfully generalized the discrimination task once the no-go stimulus was introduced. In contrast, mice that underwent a rule-reversing switch with RL activation responded with a significantly longer delay, reflecting the conflict with their prior associations regarding the new go stimulus. These mice performed at chance level once the no-go stimulus was introduced.

Additionally, we conducted new experiments to compare these results with optogenetic stimulations of AL (belonging to the ventral areas). Unexpectedly, activation of AL failed to evoke behavioral responses, even with stronger blue light intensity or when the pattern size matched the one used for RL. After the modality switch, we trained these mice for over seven days but observed no signs of opto-triggered licking responses.

When comparing these results with the time required for mice to respond to optogenetic stimulations in other conditions, we concluded that mice could not generalize sensorimotor associations learned with tactile stimulations to direct activation of AL. We believe that

stimulations of AL fail to elicit responses, at least in part, due to its weak projections to motor regions compared to area RL (see Wang et al., 2012; DOI: 10.1523/JNEUROSCI.6063-11.2012). This finding supports our original conclusion that RL, but not AL, is sufficient to generalize the task.

These results are now presented in new Figure 7, and we have also improved the description of the optogenetic experiments at pages 27-28, lines 635-662 to include a detailed analysis of habituation to optogenetic stimulations.

The neural network model fits this data well, but it is not clear to me if the model is informative in a more general sense. What do we learn from it that we did not learn from the data? How can it be applied to form a hypothesis that was not possible using the data alone? Please give a bit more information in this regard.

We believe that the model provides valuable insights into the mechanisms underlying cross-modal generalization and the necessary circuit organization to support it. In the revised manuscript, we have performed additional simulations to illustrate how differences in connectivity patterns between ventral and dorsal areas can explain their distinct contributions to cross-modal generalization. Specifically, the model confirms that strong feedback projections from RL (but not AL) to primary sensory areas, as well as strong projections from RL (but not AL) to motor cortices, are critical for enabling generalization. These connectivity patterns align with published data, highlighting an important anatomo-functional distinction between visuo-tactile areas with seemingly comparable properties. These findings are now discussed in detail at page 28-29, lines 670-689 and presented in Figure 8.

Furthermore, the model provides mechanistic insights into the involved circuit organization and synaptic plasticity required for generalized sensorimotor learning, offering avenues for experimental validation in future studies. These hypotheses extend beyond the data alone, demonstrating the broader applicability and utility of the model as discussed on page 31, lines 733-736.

Reviewer #2 (Remarks on code availability):

I do not believe the code is available yet. It looks like it will be upon publication. The authors' state "The complete data set and Matlab analysis code will be made freely available at the CERN database, Zenodo." This may or may not be a usable resource for the community and the inclusion of instructions for how to run the code should be addressed. Also, it is helpful to have an actual link. It looks like the code for the model will also be available upon publication here <https://github.com/comp-neural-circuits> but until is it, I cannot asses its usability.

We thank the reviewer; the dataset and Matlab code for this study are freely accessible on Zenodo at <https://doi.org/10.5281/zenodo.14712478>. The Python code for the simulations is freely available on GitHub at <https://github.com/comp-neural-circuits/Guyoton-Matteucci-et-al-2025.git>.

Reviewer #3 (Remarks to the Author):

Guyoton, Matteucci and collaborators (GM) study cross-modal (tactile, visual) generalization.

Mice were trained to discriminate (go/no-go) between top vs. bottom tactile (T) whisker stimuli. They were then tested on discrimination between visual stimuli presented in the top vs. bottom of the visual (V) field. Animals performed the visual task without additional training, only when the association of go/no-go with top/bottom was preserved. Generalization in the reverse direction was also studied.

To study the mechanisms of cross-modal generalization, GM performed functional mapping, anatomical tracing, and optogenetic stimulation (LoF, GoF) experiments. Those experiments support the conclusion that associative area RL, where neurons exhibit marked multisensory enhancement of their responses, has a causal role in mediating cross-modal generalization.

Overall GM studied an interesting problem and used a host of powerful experimental techniques which substantially advance our understanding of how the brain performs cross-modal generalization. However, the presentation of multiple facets of the data is oversimplified and not entirely consistent with the actual findings. The subtleties of the existing data that are overlooked in the text could be important, and should be explored further based on the computational model.

We thank the reviewer for their comments. We have conducted additional experiments, analyses, and model simulations to address the issues raised.

Major points:

GM's main finding, that RL mediates cross-modal generalization, is well supported by the experiments and controls. However, the description of the behavioral experiments overstates the symmetry between V-to-T generalization and T-to-V generalization. Inspection of Extended Data Fig. 1 a-c vs. j-l suggests that behavioral generalization dynamics of $V \rightarrow T$ vs. $T \rightarrow V$ are in fact different. GM neglect to mention those differences and thus appear to be attributing them to random variability.

We agree with the reviewer that certain aspects of the functional and anatomical mappings suggest asymmetries between the connections conveying tactile information to the visual cortex and those conveying visual information to the barrel cortex. These points are addressed below. However, we believe these asymmetries are not responsible for the behavioral differences observed between V-to-T and T-to-V switches.

The reviewer is correct in noting that performance decreases slightly when a visual task is switched to a tactile task but not the other way around. We believe this discrepancy is due to an overall difference in task difficulty specific to each modality, resulting in different performance ceilings. Specifically, the overall performance of mice in the tactile task averages around 78%, while performance in the visual task averages around 85% (as shown in Fig. 2g of the revised manuscript). This difference is likely due to the lower discriminability between nearby whiskers compared to small visual stimuli, despite matching their locations.

As a result, we expect performance to remain the same or slightly increase when switching from a tactile task to a visual task. Conversely, performance is expected to decrease when switching from a visual task to a tactile task, as observed in the data. Importantly, we believe that cross-modal generalization does not display a graded effect, and mice are equally capable of generalizing in both directions while contending with modality-specific discriminability. This clarification has been added to the manuscript on pages 8-9, lines 175-186.

Importantly, the anatomical tracing and functional mapping experiments also show marked asymmetries between visual and tactile processing.

As examples, notice that:

- a. Multisensory enhancement is rather strong in S1 (similar in magnitude to RL, Fig. 2f), while V1 shows negligible multisensory enhancement.
- b. Retrograde tracing from S1, V1 shows that S1 receives significant input from V1 (this result is inconsistent with the current discussion, line 651); while the reverse is not true (Fig. 3d vs. Extended Fig. 7g). Moreover, despite having done retrograde tracing from area RL (Extended Fig. 7h), GM do not show summary data for those experiments. Such data could support or refute the notion that RL integrates somatosensory and visual information “on equal footing”.

We thank the reviewer for suggesting a more detailed quantification of the functional and anatomical differences in our data. Indeed, Figure 2 of the original manuscript shows that visual stimuli evoke responses in S1 (while tactile stimuli barely activate V1), resulting in significant multisensory enhancement in somatosensory cortices. To properly quantify the asymmetry in how V1 and S1 drive RL, as suggested by the reviewer, we conducted additional anatomical mapping experiments using Cholera Toxin Subunit-B injected in RL and performed confocal imaging to precisely assess the neuronal density of RL-projecting neurons in V1 compared to S1. This new analysis, now presented in Extended Data Figure 8k, revealed that neurons projecting to RL are approximately 1.3 times denser in V1 than in S1, confirming an asymmetry in the information flow. This finding aligns with the functional responses we observed using wide-field imaging. We have described these new analyses in the main text on pages 15-16, lines 355-360.

Thus, multiple facets of GM's data (behavior, functional mapping, anatomical tracing) are inconsistent with symmetric integration of V and T stimuli leading to multisensory generalization. My opinion is that, given the substantial amount of experimental work that has already been performed in this study, experimentally exploring those asymmetries further goes beyond the scope of the paper. However, the computational network model in the current version – which is defined to be completely symmetric – is not consistent with the empirical findings.

We adapted the network model to account for the asymmetries observed in our data and performed a systematic investigation of their role in cross-modal generalization. In line with our behavioral results, we found that moderate asymmetries, like those reported in the experimental data, do not have a significant impact on cross-modal generalization. These new simulations and their corresponding results are now included in Extended Data Figure 14f. Additionally, we have discussed these findings in the main text on page 34, lines 810-815.

In summary, (a) The aforementioned asymmetries should be discussed explicitly; (b) The projections from S1 and V1 to RL should be quantified based on the retrograde tracing, or at least GM should explain why such quantification is not feasible; and (c) The model should be used to test whether those asymmetries (anatomical/functional/behavioral) are consistent with each other.

We believe that the new experimental data, computational simulations, and corresponding analyses confirm the existence of asymmetries in the information flow between visual and

somatosensory cortices; however, these asymmetries do not significantly affect how mice generalize learning across these modalities.

Minor points:

From the text, it is not clear what the word “learning” refers to. In the context of the results in the paper, it could refer to learning to associate top/bottom with go/no-go, or learning to generalize visual with tactile stimuli. GM should use more precise language emphasizing they refer to the former.

Learning in this context refers to the ability of mice to adapt their behavior to achieve a specific goal, which in this case involves maximizing their reward rate by responding reliably to go stimuli and not to no-go stimuli.

In the revised manuscript, we have added information regarding the training of the initial task before the modality switch (Extended Data Figure 2 of the revised manuscript). This addition better describes the process of learning the initial task and how this learned behavior is generalized to new conditions or relearned to form new associations to improve performance. Furthermore, this is substantiated by the computational model, where the learning process is explicitly modeled as changes in synaptic weights between sensory and decisional areas. These clarifications have been added to the text on page 31, lines 733-736.

GM refer to the task as involving “conscious perception”. However, no experiments were done to suggest that consciousness has anything to do with the cross-modal generalization studied in the current manuscript. For example, it could be that the neural activity mediating this generalization is also present during sleep. I believe the word conscious should be removed.

We have removed the word "conscious" from the revised manuscript to avoid any implications that consciousness is necessarily involved in the cross-modal generalization studied.

In Fig. 1e,g, Fig. 6c,d (and associated Extended Data Figures), the points corresponding to sessions -1 and 1 are connected. This is confusing, because it makes it seem as if the change in lick probability occurs before the modality switch. Those points should not be connected or connected with a dashed line.

We have used dashed lines in the revised manuscript.

In Extended Data Fig. 8g,h, the fields of view do not span the entirety of the cranial window. Areas not within a field of view should be indicated by a different color from areas that are covered yet have low density of imaged/responsive neurons. Currently one cannot tell if dark blue means no coverage or low density.

In the revised manuscript, we have updated the panel (new Extended Data Fig. 9h) to use white to indicate areas outside the fields of view, where the fraction of responsive neurons is not measured. For new Extended Data Fig. 9g, we retained the same colormap, as it represents the density of imaged neurons, which is expected to be zero outside the fields of view.

The methods section and Extended Data Figures do not make it clear enough what percentage of neurons in area RL are affected by the optogenetic manipulation or what their response profile looks like. Providing those data will strengthen the validity of the experiments.

Based on the two-photon dataset and the light patterns used to stimulate RL, we estimated the percentage of visuo-tactile neurons with generalizing properties (as defined in Figure 5f of the revised manuscript) that were stimulated after the modality switch. We estimated that approximately 65% of these neurons were activated by the blue light, with roughly 40% located in the bottom region and 25% in the top region. These estimates have been clarified and added to the text on pages 25-26, lines 604-608 of the revised manuscript.

As GM say, the results in Fig. 5e,f suggest that area RL has a unique role in multisensory generalization, compared for example with area AL. However the description of Fig. 5h,j overstates the degree to which this is confirmed by the silencing experiments. Indeed, comparing Fig. 5h to Fig. 5j shows rather small differences in effect size, even though the spatial extent of silencing in RL is larger than in AL (compare Fig. 5g,i). The authors should tone down the text describing those results, leaving the door open to the possibility that AL also plays an important role in this computation.

Our analysis based on TeNT-P2A-GFP, which specifically selected expression covering RL and AL, revealed a complete lack of generalization for RL inactivation and a transient reduction in performance for AL. It is important to note that this classification did not initially account for variability in injection site and spread. To address this, we performed additional analysis in the original manuscript, demonstrating that only GFP overlap with area RL shows correlates with impairment in generalization (Figure 6e-f of the revised manuscript). We emphasized the strength of these analysis in the revised manuscript at page 22, lines 516-519.

We acknowledge the reviewer's observation that AL inactivation slightly impairs performance after modality switch. A possible explanation is imprecision in estimating the expression pattern. Specifically, deeper-layer spread of expression (see image below), undetectable from the surface, could sometimes happen and affect nearby areas without being identified through wide-field imaging. This limitation, along with the limitations of using GFP expression in wide-field imaging to define the inactivated circuit, is now explicitly discussed in the revised manuscript at page 24, lines 569-572.

Figure. An example of a coronal brain slice showing the spread of TeNT-P2A-GFP injection within deep layers to nearby areas.

In lines 662-666, GM discuss how surround suppression of incongruent stimuli may explain multisensory enhancement. Since multisensory enhancement only occurs when stimuli are congruent, wouldn't surround suppression explain multisensory suppression and not enhancement?

The modulation pattern that we observed depended on the combination of visual and tactile stimuli, with enhancement occurring when they were spatially congruent and suppression when they were not. This implies that a nonlinear mechanism shapes the response of these neurons and cannot be explained by a simple divisive/multiplicative or subtractive/additive operation. We suggest that surround suppression could play a role in reducing neuronal firing for incongruent visuo-tactile combinations, while a separate mechanism could specifically enhance responses to congruent combinations. We have clarified this point in the revised text on page 35, lines 827-831.

Tactile stimuli are the movement of either B2 or C2 whiskers, while preferred whisker position and tuning are reported as a continuous measurement. The methods section does not include an explanation of how the preferred position was computed.

The preferred whisker position was computed as a normalized difference between the responses for the two whiskers. This has now been clarified in the methods section on pages 60-61, lines 1433-1439 of the revised manuscript.

Since each session has a different number of trials, it would be helpful to show in the Extended Data Figures (e.g., Extended Data Fig. 1b,c,e,f,h,i,k,l) the range of number of trials in each session. This would be important if, for example, the number of trials after a switch is different from the typical number of trials in other sessions.

I could not find information describing how session length was determined, though from Fig. 1 it is evident that the number of trials in each session was varied. Moreover, what was the time interval between sessions? This is important since there is a drop in lick rate in the last trials of the sessions shown in Fig. 1d,f. Does this arise from quenched thirst? If so, does the interval between sessions ensure similar thirst levels at the beginning of each session?

In the revised manuscript, we now include a comparison of the number of sessions, performance, weight loss, and session duration before modality switches across conditions. This analysis in Extended Data Fig. 2 demonstrates that mice had a comparable level of experience and motivation before being exposed to any switch. These findings are discussed on page 7, lines 140-146 and on page 9, lines 186-189.

During each session, mice were free to perform the task until their thirst was quenched, at which point they stopped performing. Session duration was measured as the number of trials during which the mouse engaged in the task. This was quantified as the trial following peak performance where the conditional lick probability for Go trials dropped below 66% of its maximum value. Mice were trained daily with a constant interval of 24 hours between sessions. As such, they always concluded a session with a drop in lick rate before waiting another 24 hours to be exposed to the task again. These clarifications have been added to the methods section on page 65, lines 1537-1539.

Point-by-point response to the reviewers' comments

Reviewer #1 (Remarks to the Author):

The authors have addressed most of my comments. In particular the updated model (Fig. 8) and the added gain-of-function experiments focused on AL (fig. 7) clarify some key aspects. Nevertheless, I was not fully satisfied by the updated loss-of-function experiments (Fig. 8). I believe that the authors should further improve this part of the manuscript. Below I outline my remaining concerns:

We thank the reviewer for their consideration of our manuscript and for their comments. In response, we performed additional statistical analyses and revised the text to clarify remaining points.

1) The authors' claim that inactivation of area RL but not AL impairs performance after modality switch is based on the finding that inactivation of RL (but not AL) induces chance performance. However, this does not seem to be quantified in Fig. 6h,j. The analysis is mentioned in the legend and main text, but the lack of asterisks in the barplots indicates no difference from chance. However, I assume this was a mistake. I also wonder if the authors might compare performance after RL vs AL inactivation. Although not done in a paired setting, this comparison would further strengthen their claims.

We appreciate the reviewer's concern regarding the absence of asterisks in Figure 6h. As noted in the figure legend (line 558 : "Blank: Not significant"), bars without asterisks denote a non-significant difference from chance ($p > 0.05$). The lack of asterisks in these plots is therefore not a mistake.

The effect of TeNT expression on cross-modal generalization depending on the targeted area is quantified in several different ways: (i) the reverse correlation map displayed in Fig. 6f, (ii) the forward correlation map displayed in Fig. 6e (iii) Statistical tests around switches for performance difference from chance or lack thereof mentioned above (Fig 6h,j). Nonetheless, we agree with the reviewer that testing directly the difference in performance after the modality switch between the AL and RL cohorts could further strengthen our claims. We performed additional statistical tests by comparing the average performance after modality switch between the AL and RL TeNT-injected groups of mice over the same time window displayed in Fig. 6j, reporting a significant difference (two-sided, two samples, unpaired, t-test, $n_{RL}=8$ and $n_{AL}=7$, $p = 0.036$). We also performed a similar quantification for the detection performance (two-sided, two sample, unpaired, t-test, $n_{RL}=8$ and $n_{AL}=7$, $p = 0.046$). We now report these tests in the revised manuscript at lines 572-576. Together, these analyses reinforce that RL—unlike AL—inactivation significantly impairs cross-modal generalization.

2) The authors suggest that the reduced performance observed following AL inactivation might stem from an expression pattern in deep layers extending to AL. However, a more straightforward explanation might simply be that AL also plays a similar - albeit weaker - than RL. I recommend that the authors expand the discussion of possible confounds and alternative

explanation, and give them a more prominent role in the Discussion (now this part is in the Results section, at lines 569-572).

We respectfully maintain that our data support a predominant role for RL over AL in cross-modal generalization. Both TeNT inactivation and optogenetic experiments yielded clearly different outcomes depending on whether RL or AL was targeted, and the computational model further aligns with RL's stronger connectivity to sensory and motor areas. We agree with the reviewer that AL may indeed contribute, albeit more modestly, to visuo-tactile integration—an interpretation now made more explicit in our revised text (lines 576–579). We have therefore rephrased our examination of potential confounds, while also emphasizing that AL's putative role does not negate the clear, experimentally demonstrated primacy of RL in driving cross-modal generalization. To preserve the manuscript's logical flow, we have opted to keep this additional point in the Results section. We believe that providing these technical clarifications in the Results section maintains the manuscript's clarity and coherence, ensuring that all relevant methodological details are presented in context.

3) The authors mention that there was no correlation between staining in V1 and S1 and a behavioral effect, but this is not shown. The authors should provide such data.

The correlation values (Pearson coefficients) between cross-modal generalization impairment and the overlap between TeNT expression patches in each relevant cortical area are presented as a color map in Fig. 6e (i.e., the "forward correlation map"). To keep the manuscript concise, given the already extensive number and size of supplementary items, we decided to display individual data points only for AL and RL in Extended Data Fig. 12g. However, in response to the reviewer's request, we provide below a figure with individual datapoints for V1 and S1 correlations. V1 exhibits a weak correlation with behavioral impairment, while S1 shows a slightly higher, yet non-significant, correlation (as indicated by its bright red color in Fig. 6e). In the revised manuscript, we now refer to the corresponding panel in the figure (line 528). This further confirms that RL remains the only area where TeNT expression overlap robustly correlates with cross-modal generalization deficits.

Figure. Correlation between the level of TeNT-P2A-GFP overlap with V1 and S1 cortical areas and the corresponding drop in discrimination performance after the modality switch. Performance changes were computed over three consecutive sessions post-switch (N=22 mice). Pearson correlation coefficients: V1 = -0.25, $p = 0.254$; S1 = -0.39, $p = 0.074$.

4) Overall, in the rebuttal the authors mention that the injection sites are not specifically confined to AL and RL. Correctly, they talk of a ventral (LI/AL) and a dorsal (RL/A) site. However, this careful description is not evident in the discussion. In order to make the manuscript better comparable to other studies, in which areas such as A and AM (but also RL) are often referred to as posterior parietal cortex, I would recommend that the authors revise their discussion to more carefully indicate which results specifically hold for RL and AL, and which should instead be assigned to broader subdivision of the mouse posterior association cortex.

We appreciate the reviewer's concern regarding the varying nomenclature used to describe rodent PPC and its subregions. In our study, we rely on a primarily "visual" nomenclature (e.g., "AL," "RL") because of our functional approach involving retinotopic mapping. Nonetheless, we acknowledge that RL is commonly considered a subdivision of rodent PPC, consistent with prior anatomical and functional reviews (Hovde et al. 2018, DOI: 10.1111/ejn.14280; Lyamzin et al. 2019, DOI: 10.1016/j.neures.2018.10.008). To improve cross-comparisons with existing literature, we explicitly note that other works referring to rodent PPC likely focus on regions overlapping with our targeted "A" and "RL" fields. We have updated lines 783–787 and 860 to reflect these points, while also noting that a more detailed analysis of the broader PPC nomenclature is beyond the scope of our current manuscript.

Reviewer #2 (Remarks to the Author):

The authors have sufficiently addressed all concerns, and I highly recommend this manuscript for publication. This research on cross-modal learning and generalization is impactful and extremely valuable to our understanding of mechanisms of cognition.

We thank the reviewer for their consideration of our manuscript.

Reviewer #3 (Remarks to the Author):

The revised version adequately addresses my comments.

We thank the reviewer for their consideration of our manuscript and for their additional comments that we addressed below.

Please see below a few remaining minor points:

1) The results of the new gain of functions experiments in AL (i.e. AL activation contrary to RL is not sufficient for generalization), together with the overlap correlation analyses, justify the

authors emphasis on RL's contribution to cross-modal generalization. I still believe however that the authors can be a little more cautious. Perhaps describe RL's role as "major", "outsized", "dominant", instead of "unique".

In the revised manuscript, we explicitly acknowledge the possibility of a minor contribution of AL to cross-modal generalization at lines 576-579, where the TeNT experiments are discussed. Moreover, in line with the reviewer's recommendation, we have replaced the term "unique" with "predominant" in the discussion (at line 797) to reflect our conclusion that RL plays a predominant—but not necessarily exclusive—role.

2) The differential levels of task difficulty for visual and tactile modalities could potentially explain the different levels of generalization in the $T \rightarrow V$ and $V \rightarrow T$ conditions. However, that still does not rule out the potential contribution of the asymmetry in the connections between S1 and V1. Relatedly, I find it strange that the model shows no functional asymmetry for any of the structural asymmetry values explored (light blue lines in Extended Data Fig. 14e,f). Here too I think the authors should be a bit more cautious and mention the possibility that asymmetry in connectivity contributes to asymmetry in generalization, especially since those asymmetries are "in the same direction".

We propose that the most direct contributor to the differences in performance after a **rule-preserving** switch is the intrinsic difference in how well mice can discriminate within each modality, rather than any moderate asymmetry in connectivity between S1 and V1. Our computational model (where stimuli are fully discriminable by design) achieves near-perfect performance immediately after a rule-preserving switch, suggesting that moderate asymmetries in feedforward connectivity ($V \rightarrow T$ vs. $T \rightarrow V$) alone are unlikely to drive the experimentally observed differences. In our model, cross-modal generalization emerges because sensory-evoked activity spreads from a primary area (e.g., V1) to the multimodal RL area and then reverberates, to a lesser degree, into the other primary area (e.g., S1). Decision neurons within RL (and ultimately S1) learn a correct stimulus-to-action mapping before the second modality is introduced, supported by congruent multisensory tuning. Introducing plausible anatomical asymmetry into these pathways does not impair rule-preserving transfers in either direction (as long as the network is sufficiently trained); the inherited weight structure still supports high performance.

We do, however, observe an effect of asymmetry during **rule-reversing** switches: stronger $V \rightarrow T$ connectivity can slow relearning for $V \rightarrow T$ transitions (due to more widespread and larger magnitude changes required in the weight structure to unlearn the previous association) compared to $T \rightarrow V$ transitions (due to fewer and lower magnitude weight adjustments required). We discuss these simulations more extensively in the revised manuscript (lines 744–755) to clarify that while connectivity asymmetry can affect relearning rates, the primary driver of performance differences in rule-preserving switches likely remains the inherent discriminability of each modality.

3) When reading the revised manuscript, I noticed that the definition of "detection performance" is rather difficult to make sense of. The readability of the paper will benefit if this definition is unpacked.

We thank the reviewer for highlighting the ambiguity around "detection performance". We have now included a clearer, more accessible explanation in the main text (lines 109–112) in

addition to the description in the methods section (lines 1575–1577). We hope this additional detail will aid readers less familiar with the technical aspects of our analysis.

Reviewer #4 (Remarks to the Author):

We thank the reviewer for their contribution to the review process.